# TimeSeed: Effective Time Series Forecasting with Sparse Endogenous Variables

## Abstract

Time series forecasting is widely applied across various domains. In real-world applications, there are many scenarios where endogenous variables are missing. Recent studies show that incorporating exogenous variables can significantly enhance the predictive accuracy of endogenous variables. However, the lack of a complete historical context introduces significant uncertainty in temporal dependence capture, particularly in systems characterized by non-stationary behavior. To address these challenges, we propose TimeSeed, specifically designed for scenarios with sparsely observed endogenous variables. Technically, TimeSeed reconstructs l sufficient endogenous series from both complete exogenous series and sparsely observed endogenous series, utilizing two types of data to extract stable information. Building on this foundation, we effectively transform the challenging original prediction task into a sequence-based prediction task. Moreover, TimeSeed is built entirely upon linear layers, which significantly reduces computational costs. Experiments conducted on seven real-world datasets demonstrate that TimeSeed consistently outperforms state-of-the-art models in forecasting accuracy, achieving an average reduction of 13.01% in MSE and 7.54% in MAE, with a model size of only 0.19M parameters. Code is available at this repository: `https://anonymous.4open.science/r/Alistair-7`.

## 1 Introduction

Nowadays, time series forecasting has become an important tool widely applied in various domains. However, in many real-world scenarios, endogenous variables are often sparsely observed, as illustrated in Figure 1 (a), encompassing applications such as weather forecasting (Ren et al., 2021; Lin et al., 2022; Lam et al., 2023), industrial forecasting (Weron, 2014; Alfares & Nazeeruddin, 2002), and battery life prediction (Sulzer et al., 2021; Fei et al., 2021).

Recent studies have demonstrated that incorporating the influence of *exogenous variables*(Huang et al., 2025; Pandit et al., 2023; Lu et al., 2024) can substantially enhance the predictive performance of *endogenous variables*(Motrenko et al., 2016). This enhancement is primarily attributed to the strong correlations between exogenous and endogenous variables, as illustrated in Figure 1 (b). Gradually, forecasting with exogenous variables (Gianfreda & Grossi, 2012) has emerged as a new paradigm. However, in sparse forecasting scenarios, this paradigm may become ineffective due to the absence of target information and the rigidity of the input structure.

To tackle such complex scenarios, it is essential to develop methods that leverage exogenous information and sparse endogenous observations for prediction. However, the main challenges stem from the following three aspects: **(1) Context Incompleteness:** The substantial absence of historical context for the endogenous variable leads to high uncertainty in causal discovery, especially in systems exhibiting non-stationary behavior (Moritz & Bartz-Beielstein, 2017). **(2) Instable Dependencies:** Sparse observational data fail to reveal dependency structures within historical time series, making it difficult for models to capture trends and dynamic patterns (Liu et al., 2022b). **(3) Uncontrolled Anomalies:** Relying on sparse endogenous observations, especially when they are outliers, may exacerbate prediction biases (Su et al., 2019).

To fill this gap, we propose TimeSeed, which reconstructs historical endogenous sequences from both endogenous and exogenous perspectives, maximally exploiting the potential of forecasting under sparse observations. Technically, we leverage the physical similarity (Huang et al., 2025;

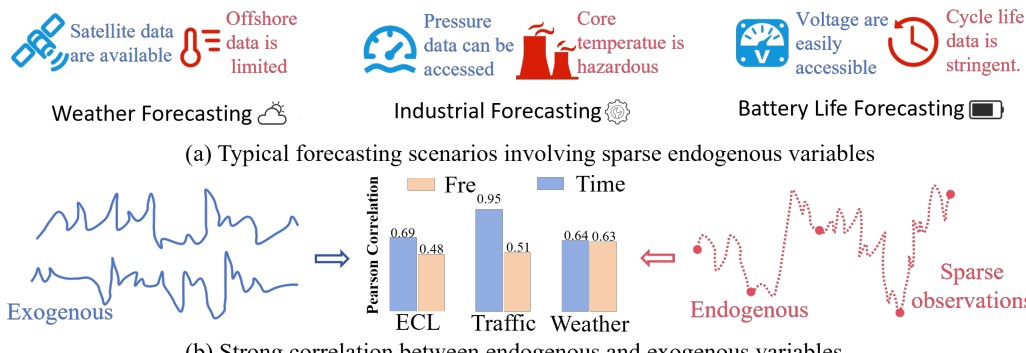

(a) Typical forecasting scenarios involving sparse endogenous variables

(b) Strong correlation between endogenous and exogenous variables

Figure 1: Scenario analysis: (a): Common real-world scenarios of sparse forecasting (b): Correlation analysis between exogenous and endogenous variables in both frequency and time domains.

Pandit et al., 2023; Lu et al., 2024) between exogenous and endogenous variables by extracting more stable sequential features from exogenous sequences that are homogeneous to the endogenous variable. Furthermore, to further enhance reconstruction stability, we propose the Adaptive Scale Reconstructor, which constructs multi-resolution representations of sparse endogenous sequences and adaptively supplements the reconstruction of the endogenous context. By reconstructing the historical context, we transform the challenging original prediction task into a sequence-based prediction task, thereby significantly reducing the forecasting difficulty. Besides, TimeSeed is built entirely on linear layers, which greatly reduces computational cost. We conduct extensive experiments on seven real-world datasets, and the results demonstrate that the proposed model achieves outstanding performance in terms of both MAE and MSE. This confirms that it can effectively utilize limited data to produce highly accurate forecasts, even under conditions of data scarcity. The main contributions can be summarized as follows:

- We propose a new prediction paradigm that relies exclusively on exogenous variables and sparse observations of the endogenous variable to forecast its future values. This effectively addresses the challenge of limited historical data for endogenous variables.

- We propose TimeSeed, a lightweight model that leverages dense exogenous and sparse endogenous sequences within a two-stage paradigm of context reconstruction and hierarchical prediction. Endogenous periodic and trend components are captured via Time Domain Aggregator (TDA) and Frequency Domain Aggregator (FDA), and refined with an Adaptive Scale Reconstructor (ASR), thereby enabling more accurate forecasts.

- We conduct comprehensive experiments on seven real-world time series forecasting datasets. Our model achieves an average reduction of 13.01% in MSE and 7.54% in MAE, with only 0.19M parameters, demonstrating its ability to significantly enhance forecasting accuracy in data-sparse settings while maintaining a compact architecture.

## 2 RELATED WORK

Exogenous variables, as key factors in improving the accuracy of endogenous variable prediction, are receiving increasing attention (Tayal et al., 2024). In traditional statistical methods, ARIMAX (Williams, 2001) has been widely used across various fields, while SARIMAX (Vagropoulos et al., 2016) further introduces radiation forecasting as an exogenous variable to enhance the accuracy of photovoltaic power generation prediction. In recent years, with the advancement of computing power and deep learning techniques, researchers have proposed various enhanced models that integrate exogenous variables. TiDE (Das et al., 2023) constructs an MLP-based encoder-decoder architecture, integrating exogenous information through feature projection and a temporal decoder. TimeXer (Wang et al., 2024b) is the first to empower the Transformer with the ability to process exogenous variables, establishing a bridge between endogenous and exogenous information through an interaction mechanism between patch-level endogenous representations and variable-level exogenous representations. In addition, NBEATSx (Olivares et al., 2023) combines neural basis functions with exogenous variables to effectively enhance power price forecasting performance.

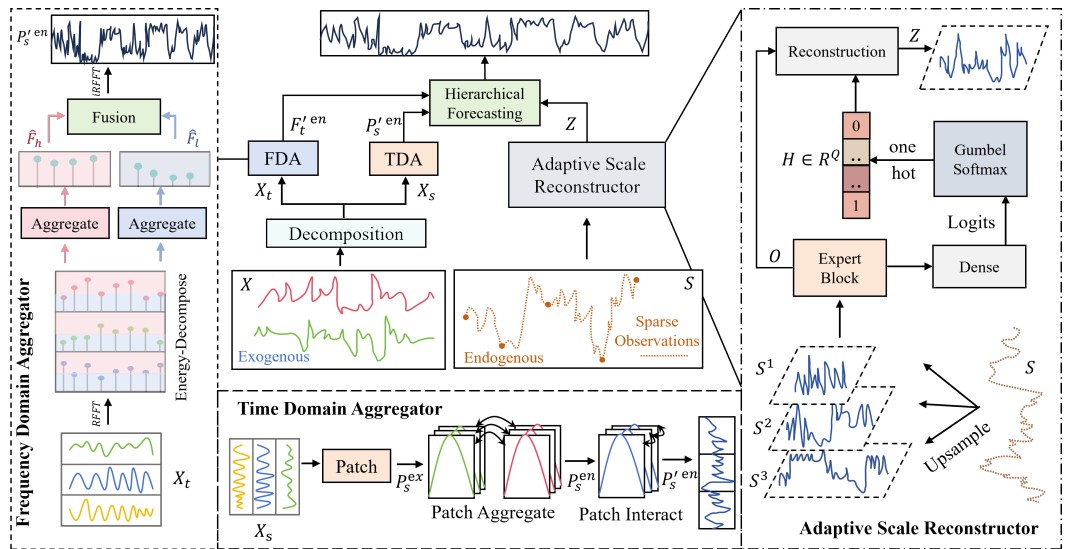

Figure 2: Overall architecture of TimeSeed, consisting of Frequency Domain Aggregator, Time Domain Aggregator, and Adaptive Scale Reconstructor, designed for sparse endogenous variables.

However, these models overly rely on exogenous-to-endogenous mapping, which becomes unreliable under sparse endogenous observations due to scale misalignment, leading to degraded predictive performance. Therefore, it is necessary to explore both sparse endogenous-to-endogenous and exogenous-to-endogenous perspectives to offset information loss and improve endogenous prediction accuracy.

# 3 METHOD

## 3.1 PROBLEM SETTINGS

Unlike traditional time series prediction, we rely only on sparse endogenous series $S$ as auxiliary information rather than full endogenous inputs. Using less endogenous variables for prediction means less information is available. Given an exogenous variable time series $X = \{x_{1:T}^{(1)}, x_{1:T}^{(2)}, ..., x_{1:T}^{(N)}\} \in \mathbb{R}^{T \times N}$ and sparse endogenous variable series $S \in \mathbb{R}^{T^{en}}$, where $x_{1:T}^{(i)}$ represents the $i$-th exogenous variable, $T$ represents the length of the lookback window, $T^{en} \ll T$ indicates the length of the sparse endogenous sequence, and $N$ represents the number of exogenous variables, the task goal is to predict the future multi-step endogenous time series $Y = \{y_{T+1}, y_{T+2}, ..., y_{T+L}\} \in \mathbb{R}^{L \times 1}$. Here, $L$ represents the number of future time steps to predict. The overall process can be described as a function mapping: $f(X, S) \to Y$.

## 3.2 STRUCTURE OVERVIEW

Since the absence of endogenous variables can substantially degrade predictive performance, our motivation is to transform the inherently complex sparse forecasting task into a sequence-based prediction task by reconstructing historical endogenous sequences. This multi-stage problem decomposition approach effectively enhances the robustness of prediction. Specifically, the reconstruction leverages the intrinsic consistency between endogenous and exogenous variables by aligning with the trends (Figure 2, left) and periodic patterns (Figure 2, middle-bottom) of exogenous variables, as well as by utilizing the sparse endogenous sequence itself (Figure 2, right).

As illustrated in Figure 2 (middle-top), our proposed model accepts exogenous sequences X and sparse endogenous observations S as inputs. Following mainstream decomposition-based models (Wu et al., 2021; Wang et al., 2023; Zhou et al., 2022b), we use a moving average method, denoted as Decomposition$(X; o)$, to downsample the input exogenous sequences $X$, yielding the separated trend component $X_t \in \mathbb{R}^{T \times N}$ and periodic component $X_s \in \mathbb{R}^{T \times N}$, where $o$ denotes the kernel size of average pooling. For these two distinct components extracted from the exogenous sequences, we

design the TDA and FDA blocks to reconstruct trend and periodic components, respectively. For the sparse endogenous observations, we propose the ASR to perform multi-scale reconstruction. Finally, the reconstruction outputs of these three modules are integrated via the Hierarchical Forecasting module to generate the final prediction results.

### 3.3 TIME DOMAIN AGGREGATOR

For different variables, similar time series patterns exist within the same period. Even in cases where endogenous variables are missing, the periodic of the endogenous variable can be restored by learning the periodicity of other variables. Based on this observation, the TDA is designed to learn the periodic features of endogenous variables by leveraging exogenous variables. Specifically, for the periodic term, we first apply the patch operation to separate the features of each patch:

$$P_s^{ex} = \text{Patch}(X_s, stride) \tag{1}$$

Here, $\text{Patch}(\cdot)$ represents the sub-patch operation on historical exogenous variables. $P_s^{ex} \in \mathbb{R}^{P \times C \times N}$ is output of $\text{Patch}(\cdot)$. $C$ represents the length of the patch, and $P = \frac{T-C}{stride} + 1$ represents the number of patches. More detailed analysis can be found in the Appendix L.

Then we aggregate the same period of different variables. Thus, a mapping from the period of the exogenous variable to the period of the endogenous variable is established:

$$P_s^{en} = \text{Patch-Agg}(P_s^{ex}) \tag{2}$$

$\text{Patch-Agg}(\cdot)$ is implemented through a linear layer along the variable axis. $P_s^{en} \in \mathbb{R}^{C \times P}$ is the aggregated one-variable feature, which could be expressed as the reconstructed endogenous variable.

Each variable exhibits a continuous time series pattern within the same period. We obtain the future period changes of endogenous variables by aggregating the same phases in the period terms of the endogenous variables during different periods:

$$P_s^{'en} = \text{Patch-Interact}(\text{Transpose}(P_s^{en})) \tag{3}$$

$\text{Patch-Interact}(\cdot)$ is implemented through linear layer, and $P_s^{'en} \in \mathbb{R}^{C \times P}$ is the endogenous feature after patch interaction. We merge specified dimensions to obtain the output $P_s^{'en} = \text{Reshape}(P_s^{'en}) \in \mathbb{R}^{T \times 1}$, which represents the period term for predicting endogenous variables.

### 3.4 FREQUENCY DOMAIN AGGREGATOR

It is observed that different variables share similar trend patterns, and these patterns can be reconstructed by aggregating trends across variables. Since trends are usually concentrated in the low-frequency domain, directly modeling them poses the risk of either overlooking local variations or distorting the main trend. Therefore, we perform operations in the frequency domain, where dominant trend-related components and local variation-related components are more clearly separated.

Specifically, we map trend signals to the frequency domain through real Fast Fourier Transform to more directly identify the dominant trend components:

$$F_t = \text{RFFT}(X_t) \tag{4}$$

Here $\text{RFFT}(\cdot)$ represents real Fast Fourier Transform. $F_t \in \mathbb{R}^{(T/2+1) \times N}$ is the frequency domain representation of the trend component of the exogenous variable.

We then perform secondary decomposition on the frequency-domain signals based on their amplitude, obtaining high-energy components that correspond to dominant trends and low-energy components that correspond to local fluctuations:

$$A_t = \text{Abs}(F_t) \tag{5}$$

$$F_h, F_l = \text{Energy-Decompose}(F_t, A_t, K) \tag{6}$$

Here, $\text{Abs}(\cdot)$ computes the energy, and $A_t \in \mathbb{R}^{(T/2+1) \times N}$ represents the distribution of energy. $\text{Energy-Decompose}(\cdot)$ separates $K$ high-energy and $T - K$ low-energy components according to the magnitude of the frequency amplitude by analyzing the spectrum. $F_h \in \mathbb{R}^{K \times N}$ and

$F_l \in \mathbb{R}^{(T/2+1-K) \times N}$ represent the high-energy and low-energy components within the trend of the exogenous variable, respectively.

Modeling these two components separately effectively leverages local details to refine predictions, while avoiding the domination of predictions by high-energy components. Specifically, we learn the mapping relationships between the dominant trends and local details of exogenous variables , and those of endogenous variables.

$$\hat{F}_h, \hat{F}_l = \text{AggHigh}(F_h), \text{AggLow}(F_l) \tag{7}$$

Here, $\text{AggHigh}(\cdot)$ and $\text{AggLow}(\cdot)$ are both implemented through frequency domain linear layer along the variable axis. $\hat{F}_h \in \mathbb{R}^{K \times 1}$ and $\hat{F}_l \in \mathbb{R}^{(T/2+1-K) \times 1}$ represent univariate features, which can be regarded as the high-energy and low-energy components within the trend of the endogenous variable, respectively.

Finally, through the inverse real Fast Fourier Transform, we obtain the trend of the reconstructed endogenous variable:

$$F_t^{'en} = \text{iRFFT}(\hat{F}_h + \hat{F}_l) \tag{8}$$

Here, iRFFT represents inverse real Fast Fourier Transform, and $F_t^{'en} \in \mathbb{R}^{T \times 1}$ represents the reconstructed trend term of the endogenous variable.

## 3.5 ADAPTIVE SCALE RECONSTRUCTOR

In order to fully reconstruct the endogenous variables, our proposed Adaptive Scale Reconstructor generates multi-resolution sequence representations through multi-scale upsampling. The original sequence is compressed into a broader representation space, allowing the model to automatically select the most appropriate scale based on the input features.

Specifically, we construct endogenous representations cross different resolutions $q \in \{1, 2, \ldots, Q\}$:

$$S^q = \text{Upsample}_q(S) \tag{9}$$

where $\text{Upsample}_q(\cdot)$ denotes the $q$-th upsampling operation applied to the input sequence $S$. At the $q$-th layer, the output sequence length is expanded from the original $T^{en}$ to $T^{en} \times 2^q$. This hierarchical upsampling progressively enlarges the temporal resolution, enabling the model to perform reconstructions from multiple perspectives.

$$O^q = \text{ExpertBlock}_q(S^q) \tag{10}$$

where $\text{ExpertBlock}_q(\cdot)$ the $q$-th expert module, which can be flexibly replaced with different task-specific networks, and $O^q \in \mathbb{R}^d$ represents the output of the $q$-th expert with $d$ denoting the hidden dimension. Each expert learns scale-specific feature representations, thereby effectively avoiding feature entanglement across different scales.

To enable the model to adaptively select the optimal scale according to the data characteristics, we further introduce the Gumbel Softmax (Jang et al., 2016) to optimize the scale selection process:

$$H = \text{Gumbel Softmax}(\text{Dense}(\{O^1, ..., O^Q\})) \tag{11}$$

where $\text{Dense}(\cdot)$ generates the corresponding logits, and $H \in \mathbb{R}^Q$ indicates the model's adaptive selection of resolution. After obtaining the most appropriate scale, the model performs feature reconstruction based on the selected representation:

$$Z = \text{Reconstruction}(\{O^1, ..., O^Q\}, H) \tag{12}$$

where $\text{Reconstruction}(\cdot)$ reconstructs the historical time series at the most appropriate resolution $O$ according to the selection scheme $H$, and $Z$ denotes the reconstructed sequence.

### 3.6 HIERARCHICAL FORECASTING

The final forecast is obtained by combining the reconstructed trend $F_t^{'en}$ and periodic components $P_s^{'en}$ (both derived from exogenous variables), with the reconstructed historical endogenous sequence $Z$ (derived from the sparse endogenous sequence). This combined result is then passed through the model's prediction head. The specific process is as follows:

$$\hat{Y}_{1:T} = F_t^{'en} + P_s^{'en} + Z \tag{13}$$

$$\hat{Y}_{T+1:T+L} = \text{Prediction}(\hat{Y}_{1:T}) \tag{14}$$

Here, $\text{Prediction}(\cdot)$ is implemented through a linear layer along the temporal axis, $\hat{Y}_{1:T}$ corresponds to the reconstruction of historical endogenous variables, and $\hat{Y}_{T+1:T+L}$ represents the prediction.

## 4 EXPERIMENTS

To evaluate the performance of TimeSeed under scenarios with sparse endogenous observations, we conduct extensive experiments based on a novel time series forecasting paradigm $f(X, S) \to Y$.

**Datasets** We use datasets that span multiple domains, including Energy (ETT (Zhou et al., 2021), ECL (Wu et al., 2021)), Weather (Wu et al., 2021), and Traffic (Wu et al., 2021). For dataset partitioning, we follow standard protocols (Lin et al., 2024a;b). Specifically, the ETT datasets are split into training, validation, and test sets with a ratio of 6:2:2, while the remaining datasets follow a 7:1:2 split. More details are provided in the Appendix B.

**Baselines** We compare TimeSeed with several state-of-the-art time series forecasting models. Include: Complex Transformer-based architectures: DUET (Qiu et al., 2024), TimeXer (Wang et al., 2024b), iTransformer (Liu et al., 2023), and PatchTST (Nie et al., 2022); Lightweight MLP-based models: TimeMixer (Wang et al., 2024a), FITS (Xu et al., 2023), CycleNet (Lin et al., 2024a), FilterNet (Yi et al., 2024) [1], SparseTSF (Lin et al., 2024b), and DLinear (Zeng et al., 2023).

**Implementation Details** For TimeSeed, we fix the patch length $P$ to 16, use a historical input window $T$ of 96 time steps, and $T_{en}$ is set to 4 by uniformly sampling the 96-step sequence at 24-step intervals. Forecasting performance is evaluated at horizons $L \in \{96, 192, 336, 720\}$. The number of high-energy components $K$ is set to 10, the number of resolutions $Q$ is set to 3, and ExpertBlock is implemented using a multi-layer perceptron. We upsample the sparse endogenous series and maintain the length of the exogenous sequences, ensuring compatibility with baseline inputs and enabling fair comparison across models. In addition, we unify the hyperparameters across all models and report the rerun results. More details are provided in the Appendix B.

### 4.1 MAIN RESULTS

We validate the effectiveness of TimeSeed on long-term time series forecasting tasks under sparse scenarios (sparsity ratio of 4%) across seven mainstream benchmark datasets. As shown in the experimental results in Table 1, TimeSeed achieves nearly optimal performance across all datasets. Specifically, under sparse settings, it yields an average MSE improvement of 15.04% on the ETTh1 dataset and 19.24% on the Traffic dataset, demonstrating a clear advantage over DLinear and PatchTST, which represent competitive Linear-based and Transformer-based models, respectively. Notably, as shown in the experimental results in Table 2, when endogenous variables are missing, several state-of-the-art models exhibit performance degradation, likely due to their heavy reliance on complete endogenous sequences particularly under more challenging single-point sparse forecasting scenarios. More detailed results are provided in Appendix N and G.

### 4.2 EFFECT OF SPARSITY RATIOS

Table 3 presents TimeSeed's performance under varying sparsity ratios. The results demonstrate that as the sparsity ratio increases (i.e., a higher proportion of endogenous variables), predictive accuracy improves consistently across both ETTh2 and ETTm2 datasets. Specifically, on the ETTh2

---

[1]Implemented in TexFilter and PaiFilter, respectively

Table 1: Unified hyperparameter for long-term time series forecasting results are **based on sparse endogenous variable setting**, with a 24-hour sampling interval.

| Model | TimeSeed | | DUET | | iTrans | | DLinear | | TimeXer | | TimeMixer | | PatchTST | | FITS | | CycleNet | | TexFilter | | PaiFilter | | SparseTSF | |
|---|---|---|---|---|---|---|---|---|---|---|---|---|---|---|---|---|---|---|---|---|---|---|---|---|
| Metric | MSE | MAE | MSE | MAE | MSE | MAE | MSE | MAE | MSE | MAE | MSE | MAE | MSE | MAE | MSE | MAE | MSE | MAE | MSE | MAE | MSE | MAE | MSE | MAE |
| ETTh1 | 0.096 | 0.242 | 0.159 | 0.311 | 0.186 | 0.345 | 0.113 | 0.262 | 0.444 | 0.569 | 0.162 | 0.309 | 0.228 | 0.381 | 0.142 | 0.297 | 0.116 | 0.261 | 0.144 | 0.294 | 0.152 | 0.306 | 0.158 | 0.316 |
| ETTh2 | 0.272 | 0.410 | 0.381 | 0.491 | 0.902 | 0.811 | 0.409 | 0.493 | 0.342 | 0.465 | 0.274 | 0.412 | 0.574 | 0.617 | 0.553 | 0.587 | 0.291 | 0.423 | 0.328 | 0.456 | 0.475 | 0.545 | 0.622 | 0.635 |
| ETTm1 | 0.068 | 0.194 | 0.157 | 0.303 | 0.143 | 0.282 | 0.075 | 0.205 | 0.344 | 0.499 | 0.141 | 0.305 | 0.104 | 0.245 | 0.079 | 0.216 | 0.104 | 0.246 | 0.145 | 0.294 | 0.109 | 0.254 | 0.499 | 0.466 |
| ETTm2 | 0.149 | 0.289 | 0.284 | 0.417 | 0.373 | 0.495 | 0.165 | 0.303 | 0.266 | 0.404 | 0.179 | 0.325 | 0.208 | 0.362 | 0.202 | 0.343 | 0.157 | 0.296 | 0.176 | 0.318 | 0.176 | 0.320 | 0.193 | 0.339 |
| Weather | 0.002 | 0.034 | 0.006 | 0.058 | 0.011 | 0.080 | 0.009 | 0.066 | 0.827 | 0.789 | 0.394 | 0.552 | 0.004 | 0.050 | 0.007 | 0.067 | 0.003 | 0.039 | 0.005 | 0.052 | 0.012 | 0.077 | 0.007 | 0.068 |
| ECL | 0.529 | 0.558 | 0.641 | 0.618 | 0.749 | 0.676 | 0.832 | 0.699 | 0.575 | 0.573 | 1.674 | 0.987 | 0.714 | 0.659 | 1.087 | 0.817 | 0.822 | 0.690 | 0.775 | 0.677 | 1.027 | 0.809 | 0.709 | 0.656 |
| Traffic | 0.403 | 0.433 | 0.574 | 0.561 | 0.483 | 0.493 | 0.584 | 0.542 | 1.396 | 1.028 | 1.251 | 0.828 | 0.499 | 0.509 | 0.790 | 0.672 | 0.876 | 0.678 | 0.584 | 0.548 | 0.703 | 0.619 | 1.276 | 0.886 |

Table 2: Unified hyperparameter for long-term time series forecasting results are **based on a single endogenous variable setting**, with the point selection strategy choosing the most recent time step.

| Model | TimeSeed | | DUET | | iTrans | | DLinear | | TimeXer | | TimeMixer | | PatchTST | | FITS | | CycleNet | | TexFilter | | PaiFilter | | SparseTSF | |
|---|---|---|---|---|---|---|---|---|---|---|---|---|---|---|---|---|---|---|---|---|---|---|---|---|
| Metric | MSE | MAE | MSE | MAE | MSE | MAE | MSE | MAE | MSE | MAE | MSE | MAE | MSE | MAE | MSE | MAE | MSE | MAE | MSE | MAE | MSE | MAE | MSE | MAE |
| ETTh1 | 0.101 | 0.249 | 0.189 | 0.332 | 0.179 | 0.332 | 0.116 | 0.268 | 0.380 | 0.525 | 0.538 | 0.632 | 0.168 | 0.321 | 0.123 | 0.278 | 0.343 | 0.492 | 0.125 | 0.276 | 0.121 | 0.273 | 0.168 | 0.328 |
| ETTh2 | 0.269 | 0.409 | 0.367 | 0.481 | 0.912 | 0.816 | 0.296 | 0.430 | 0.344 | 0.466 | 0.540 | 0.576 | 0.621 | 0.648 | 0.723 | 0.680 | 0.343 | 0.465 | 0.400 | 0.499 | 0.462 | 0.537 | 0.676 | 0.661 |
| ETTm1 | 0.061 | 0.189 | 0.096 | 0.237 | 0.083 | 0.223 | 0.065 | 0.196 | 0.325 | 0.489 | 0.488 | 0.566 | 0.080 | 0.218 | 0.061 | 0.190 | 0.233 | 0.400 | 0.073 | 0.206 | 0.083 | 0.222 | 0.090 | 0.231 |
| ETTm2 | 0.176 | 0.321 | 0.283 | 0.417 | 0.345 | 0.468 | 0.206 | 0.347 | 0.268 | 0.401 | 0.239 | 0.377 | 0.200 | 0.345 | 0.248 | 0.424 | 0.283 | 0.415 | 0.216 | 0.356 | 0.231 | 0.377 | 0.249 | 0.392 |
| ECL | 0.569 | 0.580 | 0.715 | 0.659 | 0.838 | 0.716 | 1.083 | 0.799 | 0.619 | 0.594 | 0.662 | 0.628 | 0.813 | 0.701 | 1.099 | 0.812 | 0.747 | 0.657 | 0.853 | 0.708 | 0.813 | 0.699 | 0.886 | 0.734 |
| Traffic | 0.482 | 0.489 | 0.585 | 0.570 | 0.505 | 0.507 | 0.602 | 0.553 | 1.549 | 1.078 | 1.405 | 1.041 | 0.477 | 0.498 | 0.916 | 0.741 | 1.555 | 1.083 | 0.763 | 0.646 | 1.027 | 0.805 | 1.597 | 0.990 |
| Weather | 0.002 | 0.034 | 0.006 | 0.059 | 0.009 | 0.073 | 0.009 | 0.074 | 0.742 | 0.737 | 2.527 | 1.406 | 0.004 | 0.048 | 0.007 | 0.065 | 0.833 | 0.789 | 0.005 | 0.056 | 0.012 | 0.078 | 0.005 | 0.059 |

dataset, increasing the sparsity ratio yields a 16.5% reduction in average MSE. On the ETTm2 dataset, we observe a similar trend with an 18.8% decrease in average MSE. These performance gains are sustained across all forecasting horizons, confirming that incorporating richer endogenous information substantially enhances forecasting capability under sparse endogenous settings.

Table 3: Forecasting performance on ETTh2 and ETTm2 under different sparsity ratios of endogenous to exogenous features (4%–50%). SR denotes sparsity ratios (endogenous : exogenous).

| SR | | 4% (default) | | 8% | | 16% | | 25% | | 33% | | 50% | |
|---|---|---|---|---|---|---|---|---|---|---|---|---|---|
| | Metric | MSE | MAE | MSE | MAE | MSE | MAE | MSE | MAE | MSE | MAE | MSE | MAE |
| ETTh2 | 96 | 0.186 | 0.336 | 0.167 | 0.318 | 0.154 | 0.304 | 0.152 | 0.303 | 0.147 | 0.297 | 0.144 | 0.293 |
| | 192 | 0.234 | 0.379 | 0.212 | 0.361 | 0.204 | 0.352 | 0.195 | 0.343 | 0.198 | 0.347 | 0.189 | 0.337 |
| | 336 | 0.288 | 0.426 | 0.267 | 0.410 | 0.265 | 0.409 | 0.250 | 0.396 | 0.249 | 0.396 | 0.241 | 0.389 |
| | 720 | 0.380 | 0.499 | 0.374 | 0.495 | 0.364 | 0.489 | 0.350 | 0.480 | 0.344 | 0.476 | 0.334 | 0.470 |
| | AVG | 0.272 | 0.410 | 0.255 | 0.396 | 0.247 | 0.388 | 0.237 | 0.381 | 0.234 | 0.379 | 0.227 | 0.372 |
| ETTm2 | 96 | 0.092 | 0.224 | 0.075 | 0.202 | 0.071 | 0.197 | 0.070 | 0.194 | 0.070 | 0.195 | 0.070 | 0.194 |
| | 192 | 0.127 | 0.269 | 0.109 | 0.249 | 0.103 | 0.240 | 0.102 | 0.239 | 0.102 | 0.238 | 0.103 | 0.239 |
| | 336 | 0.160 | 0.304 | 0.139 | 0.283 | 0.135 | 0.278 | 0.132 | 0.275 | 0.131 | 0.274 | 0.131 | 0.274 |
| | 720 | 0.217 | 0.360 | 0.192 | 0.337 | 0.183 | 0.327 | 0.184 | 0.328 | 0.183 | 0.327 | 0.182 | 0.325 |
| | AVG | 0.149 | 0.289 | 0.129 | 0.268 | 0.123 | 0.261 | 0.122 | 0.259 | 0.122 | 0.259 | 0.121 | 0.258 |

## 4.3 ABLATION STUDY

To further analyze the contribution of each component to the model's performance, we performed ablation analysis on the Time Domain Aggregator, Frequency Domain Aggregator, and Adaptive Scale Reconstructor to assess their individual impacts. As shown in Table 4, we draw the following three conclusions: (1) All modules positively contribute to the performance of TimeSeed, with improvements in MSE ranging from 5.31% to 12.30%. (2) Comparatively, the contribution of high-energy information to the prediction results is slightly lower, leading to an MSE improvement of about 5.31%. (3)Based on a two-stage decomposition and forecasting paradigm, outperforms the Direct Forecasting approach in terms of evaluation metrics. Notably, on the ETTh2 and ETTm2 datasets, TimeSeed achieves average improvements of 27.4% and 14.2% in MSE and MAE, respectively. This is because the two-stage approach decouples the sequence features, allowing the reconstruction stage to focus more on capturing trends and periodic patterns, thereby enhancing robustness(4) Furthermore, different datasets rely to varying degrees on information from the time domain, the frequency domain, and the multi-resolution reconstructions derived from sparse endogenous series. More detailes can be found in the Appendix H.

Table 4: Ablation study results. FDA denotes Frequency Domain Aggregator, ASR denotes Adaptive Scale Reconstructor and TDA denotes Time Domain Aggregator.

| Datasets | ETTh1 | | ETTm1 | | ETTh2 | | ETTm2 | |
|---|---|---|---|---|---|---|---|---|
| Metric | MSE | MAE | MSE | MAE | MSE | MAE | MSE | MAE |
| TimeSeed | **0.096** | **0.242** | **0.068** | **0.194** | **0.272** | **0.410** | **0.149** | **0.289** |
| w/o Agghigh (Eq. 7) | 0.112 | 0.242 | 0.084 | 0.215 | 0.284 | 0.420 | 0.155 | 0.295 |
| w/o Agglow (Eq. 7) | 0.110 | 0.257 | 0.077 | 0.206 | 0.285 | 0.420 | 0.159 | 0.299 |
| w/o Patch-Interact (Eq. 3) | 0.119 | 0.265 | 0.074 | 0.202 | 0.291 | 0.424 | 0.156 | 0.298 |
| w/o FDA | 0.118 | 0.265 | 0.088 | 0.222 | 0.299 | 0.432 | 0.154 | 0.293 |
| w/o TDA | 0.117 | 0.263 | 0.083 | 0.219 | 0.376 | 0.488 | 0.154 | 0.295 |
| w/o ASR | 0.099 | 0.246 | 0.082 | 0.218 | 0.279 | 0.415 | 0.176 | 0.322 |
| Direct Forecasting | 0.124 | 0.274 | 0.074 | 0.204 | 0.400 | 0.493 | 0.160 | 0.300 |

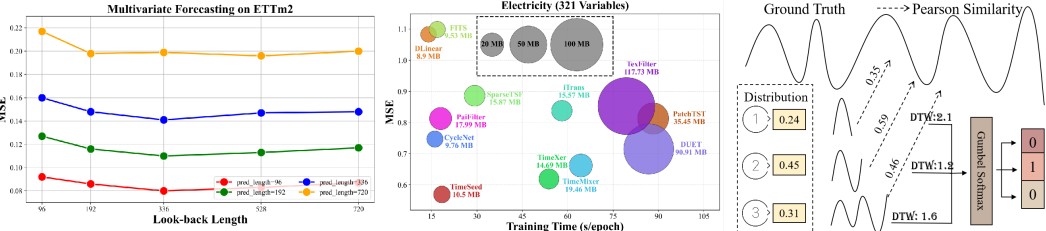

Figure 3: Analysis of the TimeSeed. Left : Performance across various prediction lengths with different look-back window sizes. Middle : Comparison of model efficiency. Right: Selection distribution of TimeSeed across multi-level sequences.

## 4.4 MODEL ANALYSIS

**Different Look-back Window Sizes** To assess robustness, we vary look-back lengths on ETTm2. Longer windows provide richer history but may cause redundancy for linear models. As shown in Figure 3 (Left), TimeSeed benefits from longer windows, with clear MSE gains at horizons 192 and 336, demonstrating stable performance across input lengths. Detailed results are in Appendix C.

**Efficiency Analysis** To assess computational efficiency, we compared TimeSeed with 11 state-of-the-art models on GPU memory and training time. Under the same settings of hidden dimension 128 and batch size 1, results in Figure 3 (Middle) demonstrate that TimeSeed achieves superior performance in both memory efficiency and predictive accuracy. It consumes only 10.5MB of memory, approximately 71.5% of TimeXer's 14.69MB. In addition, its training time is roughly 30% of that of TimeXer, further emphasizing its computational efficiency. More results are in the Appendix D.

**Case Study** To validate the effectiveness of the adaptive multi-resolution selection mechanism in the proposed ASR, we analyze a representative case from the ETTh1 dataset. As shown in Figure 3 (Right), we further report the correlation and Dynamic Time Warping (DTW) between the reconstructed sequences at each resolution and the ground truth. The second resolution exhibits the highest correlation with the ground truth, achieving a Pearson coefficient of 0.59 and a DTW of 1.2, which indicates its superiority as the most appropriate resolution. Furthermore, ASR accurately identifies this candidate sequence, thereby validating the effectiveness of its adaptive multi-resolution selection mechanism.

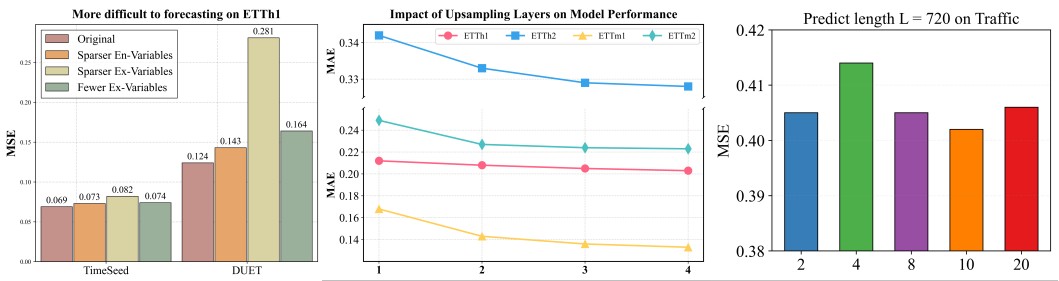

Figure 4: Sensitivity Analysis to different Data Scales, Resolutions and High-energy Components.

### 4.5 SENSITIVITY ANALYSIS

**Different Data Scale** To further explore the potential of TimeSeed under sparse settings, we design three more challenging forecasting scenarios: sparser endogenous series, fewer exogenous variables, and shorter historical exogenous series. As shown in Figure 4 (left), the performance of TimeSeed decreases by 5.80%, 18.84%, and 7.25% compared to the original setting under the three different configurations. In contrast, DUET shows a larger decline, with decreases of 15.32%, 126.61%, and 32.26%, respectively. it is evident that among these three factors, the length of the historical exogenous series has the greatest impact on forecasting performance. TimeSeed consistently maintains the best predictive performance across all cases.

**Different Resolution** To further investigate the impact of ASR under different resolution choices, we vary the number of upsampling layers from 1 to 4. As shown in Figure 4 (middle), with more upsampling layers, the number of available resolutions increases. On the ETT datasets, performance improves progressively, with a particularly significant gain when increasing the layer count from 1 to 2. Considering both performance and efficiency, we set the number of resolutions $Q$ to 3.

**Different High-energy Components** As shown in Figure 4 (Right), increasing the decomposition factor initially improves performance; however, beyond a certain point (e.g., 8), the gains plateau or even diminish. This phenomenon suggests that an appropriately chosen number of high-energy components is beneficial for model performance. Therefore, we unify $K = 10$ in our implementation for experiments. More detailed results can be found in the Appendix F.

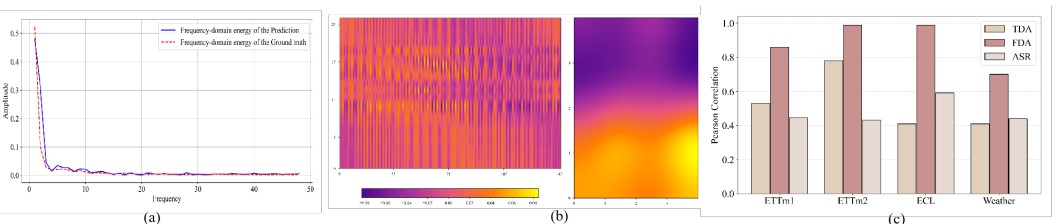

Figure 5: Reliability analysis of reconstruction. (a) Frequency spectrum of the reconstructed trend component of historical endogenous variables. (b) Visualization of the weights in the TDA. (c) Correlation between reconstruction by TDA, FDA and ASR and those of the ground truth.

### 4.6 RELIABILITY ANALYSIS OF RECONSTRUCTION

**Frequency-domain Reconstruction** Figure 5 (a) shows that the reconstructed and true trend components on ETTh1. It is clearly observable that the two curves almost completely overlap. This indicates that TimeSeed can accurately reconstruct endogenous trends from exogenous inputs, effectively capturing the relationship between exogenous and endogenous trends and validating the effectiveness of our frequency domain multi-granularity modeling. More results are in Appendix E.

**Time-domain Visualization** Figure 5 (b) shows weight heatmaps of the Time Domain Aggregator. The left panel corresponds to the implementation of $\mathrm{Patch\text{-}Agg}(\cdot)$, and the right panel to $\mathrm{Patch\text{-}Interact}(\cdot)$. The repeated horizontal purple stripes on the left indicate that $\mathrm{Patch\text{-}Agg}(\cdot)$ is sensitive to periodic features in the input sequence. In contrast, the heatmap on the right exhibits a smooth top-down gradient, suggesting that $\mathrm{Patch\text{-}Interact}(\cdot)$ effectively captures cross-period feature correlations. These distributions suggest the model adaptively emphasizes phase-aligned information, enhancing temporal structure modeling.

**Reconstruction Correlation** In Figure 5 (right), we report the Pearson correlation coefficients between the reconstructed endogenous periodic and trend components obtained by TDA and FDA and the corresponding ground truth. Overall, both TDA and FDA achieve sufficiently high reconstruction fidelity, with average correlations of 0.532 and 0.885, respectively. This further highlights the effectiveness of our two-stage decomposition and forecasting paradigm based on reconstruction.

### 4.7 MORE EXPERIMENTS

**Why not Choose Weighted Combination from All Reconstructed Resolution:** In scenarios with extremely sparse data, the quality of upsampled sequences across resolutions can differ substantially.

Table 5: Comparison of Soft and Hard choose for multi-resolution selection

| Model | TimeSeed | | TimeSeed(Soft) | |
|---|---|---|---|---|
| Metric | MSE | MAE | MSE | MAE |
| ETTh1 | **0.096** | **0.242** | 0.119 | 0.266 |
| ETTm1 | **0.068** | **0.194** | 0.092 | 0.226 |
| Traffic | **0.403** | **0.433** | 0.545 | 0.567 |
| Weather | **0.002** | **0.034** | 0.009 | 0.071 |

Soft weighted combinations (e.g., using all $O_q$) may blend high-quality signals with low-quality or noisy ones, degrading performance. In contrast, a hard-selection mechanism alleviates this issue while keeping model complexity in check and reducing overfitting. For a lightweight model with only 0.19M parameters, attending to a single resolution is both more efficient and more robust, as it encourages the model to focus on fundamental patterns rather than noise. As shown in Table 13, hard selection consistently outperforms soft weighting on all datasets, benefiting from its ability to exclude unreliable resolutions and thereby lower overfitting risk.

Table 6: Results on the sparse real-world PhysioNet dataset

| Model | TimeSeed | | DUET | | DLinear | | TimeXer | | FilterNet | | SparseTSF | |
|---|---|---|---|---|---|---|---|---|---|---|---|---|
| Metric | MSE | MAE | MSE | MAE | MSE | MAE | MSE | MAE | MSE | MAE | MSE | MAE |
| PhysioNet | **0.30** | **0.24** | 0.34 | **0.24** | 0.33 | **0.24** | 0.78 | 0.53 | 0.33 | **0.24** | 0.35 | 0.25 |

**Real-world Benchmark:** We have incorporated the real clinical dataset $PhysioNet$, where physiological variables naturally exhibit sparsity. We use reliably obtainable signals such as HR, RespRate, Temp, SysABP, DiasABP, and MAP as exogenous variables, while the more sparsely observed Glucose serves as the endogenous variable. We use the first 24 hours of observations to forecast the subsequent 24 hours. As shown in Table 6, TimeSeed still achieves the best or highly competitive performance under these genuinely sparse conditions, demonstrating the robustness and practical applicability of our method.

Table 7: Comparison between the imputation+forecasting pipeline and TimeSeed

| Model | TimeSeed | | PatchT/PatchT | | PatchT/TimeX | |
|---|---|---|---|---|---|
| Metric | MSE | MAE | MSE | MAE | MSE | MAE |
| ETTh1 | **0.096** | **0.242** | 0.153 | 0.322 | 0.388 | 0.498 |
| ETTh2 | **0.272** | **0.410** | 0.502 | 0.561 | 0.281 | 0.412 |
| ETTm1 | 0.068 | 0.194 | **0.041** | **0.196** | 0.283 | 0.428 |
| ETTm2 | **0.149** | **0.289** | 0.154 | 0.293 | 0.197 | 0.336 |

**Imputation model + forecasting model:** We first use PatchTST to impute the sparse endogenous series, and then apply another state-of-the-art forecasting model (PatchTST or TimeXer) to predict future values. As shown in Table 7, TimeSeed outperforms both combinations on most datasets. On average, relative to PatchTST/PatchTST, TimeSeed reduces MSE and MAE by about 36% and 24%, and relative to PatchTST/TimeXer, by about 26% and 30%, respectively. These gains stem from the two-stage paradigm, which decouples sequence features and enables the reconstruction stage to more effectively capture trends and periodicity, improving robustness. The limited benefit of the imputation model likely results from the extreme sparsity of the endogenous observations, while adding another imputation module increases parameter count and thus the risk of overfitting.

## 5 CONCLUSION

We propose TimeSeed, a novel prediction architecture tailored for scenarios with sparse endogenous variables. From both the endogenous and exogenous perspectives, TimeSeed can robustly reconstruct historical endogenous sequences by uncovering the periodic and trend-related relationships between exogenous and endogenous variables. Moreover, it leverages ASR to supplement the reconstructed endogenous information with signals from the sparse endogenous sequences. All experimental results show that TimeSeed achieves the best performance across all benchmarks, demonstrating its ability to deliver high-accuracy predictions even under extreme data scarcity. Furthermore, thanks to its linear-based architecture, TimeSeed exhibits excellent computational efficiency. These advantages provide a practical and effective solution to the challenge of missing data in real-world applications. In future work, we plan to explore more advanced prediction methods for more complex scenarios, such as when *endogenous variables are entirely missing*.

ETHICS STATEMENT

All authors have read and adhered to the ICLR Code of Ethics. This work does not involve human subjects, private data, or sensitive content. The research is based solely on publicly available datasets and standard benchmarks, with no foreseeable harmful societal or environmental impacts. No conflicts of interest or external sponsorships that could bias the results are involved.

REPRODUCIBILITY STATEMENT

We have made significant efforts to ensure the reproducibility of our work. The proposed model and training procedure are described in detail in Sections 3 and 4. All hyperparameters, implementation details, and evaluation protocols are provided in the appendix B. The datasets used in our experiments are publicly accessible, and we include the preprocessing steps in the supplementary material. In addition, we provide anonymized source code and instructions as supplementary materials to facilitate replication of our results.

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

# TimeSeed: Effective Time Series Forecasting with Sparse Endogenous Variables

# ————Appendix————

## CONTENTS

## A  RELATE WORK OF LIGHTWEIGHT FORECASTING

In recent years, the field of long-term time series forecasting (LTSF) (Lin et al., 2023; Zhang & Yan, 2023; Wu et al., 2022; Liu et al., 2022a; Zhou et al., 2022a; Tang & Zhang, 2025; Qiu et al., 2024) has seen a surge in lightweight models. DLinear (Zeng et al., 2023) achieves accurate forecasting using only linear layers and a decomposition strategy. CycleNet(Lin et al., 2024a) utilizes learnable parameters to simulate periodic variations across datasets, enabling plug-and-play lightweight forecasting. FITS (Xu et al., 2023) introduces a low-pass filter in the frequency domain to reduce parameter requirements, compressing the model size to approximately 10k parameters. SparseTSF(Lin et al., 2024b) decouples periodicity and trend through cross-period sparse forecasting. It first downsamples the original series using a fixed periodicity and then predicts each downsampled subsequence. MixLinear (Ma et al., 2024) further combines temporal and frequency domain feature extraction. By downsampling the series, it reduces the parameter complexity of linear models from $O(N^2)$ to $O(N)$, achieving efficient computation.

However, the above methods focus solely on lightweight modeling of the temporal characteristics of the target variable, without considering the crucial relationship between exogenous and endogenous variables (Huang et al., 2025; Das et al., 2023; Wang et al., 2024b), where a factor that is particularly important in endogenous variable prediction scenarios. Therefore, to facilitate lightweight extraction of external (exogenous) knowledge in such contexts, it is crucial to develop a compact modeling approach that captures deep correlations between endogenous and exogenous variables, enabling efficient and accurate endogenous prediction.

## B   IMPLEMENTATION DETAILS

**DataSets** We evaluated the performance of TimeSeed on seven widely used datasets. These include the Traffic (Wu et al., 2021), Weather (Wu et al., 2021), Electricity (Wu et al., 2021), and the ETT dataset (Zhou et al., 2021). Specifically, Traffic records traffic flow freeway system of the San Francisco area, collected at hourly intervals via inductive loop detectors installed on roadways, and has been collected since 2015. Weather collects 21 weather metrics from the National Weather Service (NWS), including temperature, humidity, wind speed, and barometric pressure, covering nearly 400 weather stations across the United States. This data is collected every 10 minutes. Electricity records hourly power consumption data for 321 customers. The ETT contains electrical load and oil temperature data from two substations, which are organized into four sub-datasets: ETTh1, ETTh2, ETTm1, and ETTm2, where "h" stands for hourly sampling intervals and "m" stands for 15-minute sampling intervals. For the ETT dataset, the time period spanning from July 2016 to July 2018 includes electrical load, oil temperature, and six other relevant metrics. Overall, the datasets we use cover diverse domains such as transportation, meteorology, energy, etc., with varying temporal granularities. Detailed information about these datasets is provided in Table 8.

Table 8: Comparison of dataset characteristics, including key information such as the definitions of endogenous (En.Explanation) and exogenous (Ex.Explanation), prediction horizon, sampling frequency, and dataset size (training, validation, and test sets).

| Dataset | ETTh1 | ETTh2 | ETTm1 | ETTm2 | ECL | Traffic | Weather |
|---|---|---|---|---|---|---|---|
| Ex.Explanation | Energy Load | Energy Load | Energy Load | Energy Load | Power consumption | Road Occupancy | Weather Indicators |
| En.Explanation | Oil Temperature | Oil Temperature | Oil Temperature | Oil Temperature | Power consumption | Road Occupancy | CO2-Concentration |
| Predict Length | (96,192,336,720) | (96,192,336,720) | (96,192,336,720) | (96,192,336,720) | (96,192,336,720) | (96,192,336,720) | (96,192,336,720) |
| Ex.Count | 6 | 6 | 6 | 6 | 320 | 861 | 20 |
| Sampling Frequency | 1 Hour | 1 Hour | 15 Minutes | 15 Minutes | 1 Hour | 1 Hour | 10 Minutes |
| Dataset Size | (8449,2785,2785) | (8449,2785,2785) | (34369,11425,11425) | (34369,11425,11425) | (15591,5167,5165) | (110335,3415,3415) | (31426,10445,10445) |

**Unified Hyperparameter Settings** Under our newly proposed forecasting paradigm, we fix the hyperparameters for all models and adopt the same optimization strategy to ensure fair and reproducible experiments. The detailed settings are shown in Table 9. All the experiments are implemented in PyTorch (Paszke, 2019) and conducted on a single NVIDIA 2080Ti 10GB GPU.

Table 9: Unified hyperparameter settings for all experiments. All models are optimized using the ADAM optimizer (Kingma & Ba, 2014). $K$ the number of high-energy components corresponding to those extracted by the FDA. $D_{model}$ represents the hidden dimension of the baseline model, and $D_{ff}$ is the baseline model's dimension of the hidden layer in the feed-forward layer.

| Dataset / Configurations | Model Hyper-parameter | | | Training Process | | | | |
|---|---|---|---|---|---|---|---|---|
| | $K$ | $D_{model}$ | $D_{ff}$ | Batchsize | Lr | Epoch | Early_stop | Loss |
| ETTh1 | 10 | 128 | 512 | 128 | 0.001 | 10 | 3 | MSE |
| ETTh2 | 10 | 128 | 512 | 128 | 0.001 | 10 | 3 | MSE |
| ETTm1 | 10 | 128 | 512 | 128 | 0.001 | 10 | 3 | MSE |
| ETTm2 | 10 | 128 | 512 | 128 | 0.001 | 10 | 3 | MSE |
| ECL | 10 | 128 | 512 | 4 | 0.001 | 10 | 3 | MSE |
| Traffic | 10 | 128 | 512 | 4 | 0.001 | 10 | 3 | MSE |
| Weather | 10 | 128 | 512 | 64 | 0.001 | 10 | 3 | MSE |

# C  FULL RESULTS OF DIFFERENT LOOK-BACK WINDOW SIZE

Figure 6 and 7 presents the impact of varying look-back window sizes on the prediction accuracy of TimeSeed across the ETT series datasets.

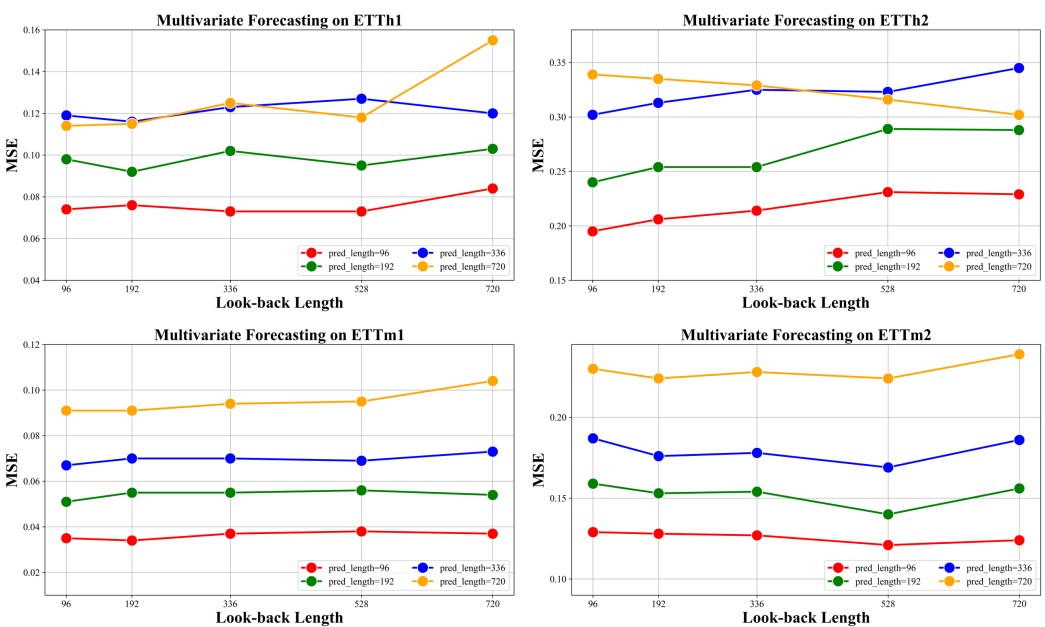

Figure 6: Performance across various prediction lengths with different look-back window sizes $T = \{96, 192, 336, 528, 720\}$ under a single endogenous setting. Each colored curve represents the performance of a specific look-back window.

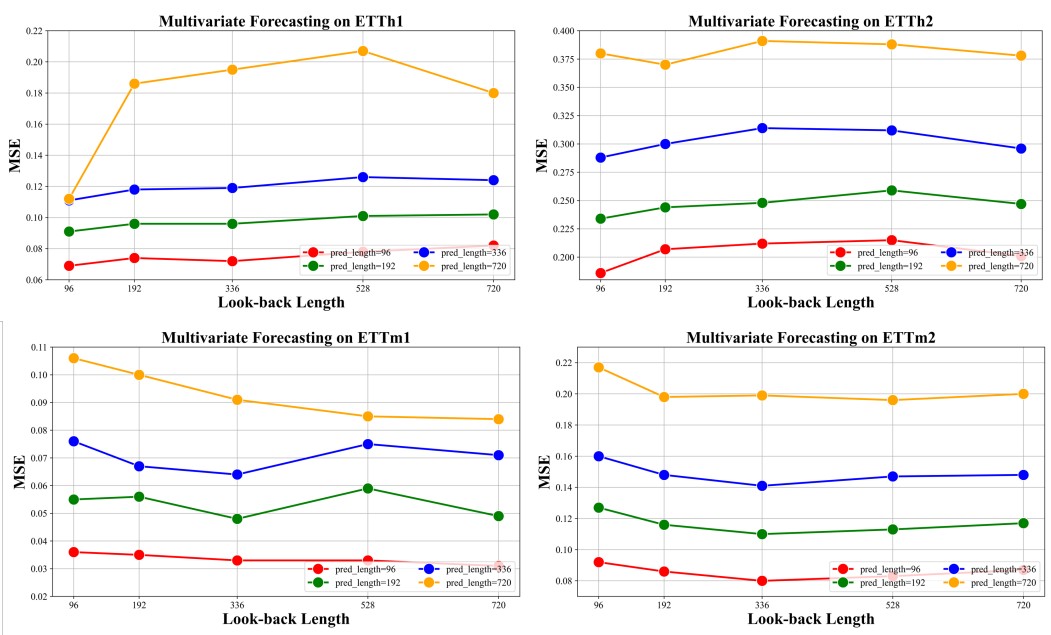

Figure 7: Performance across various prediction lengths with different look-back window sizes $T = \{96, 192, 336, 528, 720\}$ under sparse endogenous setting. Each colored curve represents the performance of a specific look-back window.

# D FULL RESULTS OF RUNTIME EFFICIENCY ANALYSIS

We report the runtime efficiency of all models across all datasets in terms of training time (s/epoch) and GPU memory usage, along with predictive performance measured by MSE. As shown in Figure 8, TimeSeed fully unleashes the potential of linear layers, achieving accurate predictions while maintaining a lightweight design.

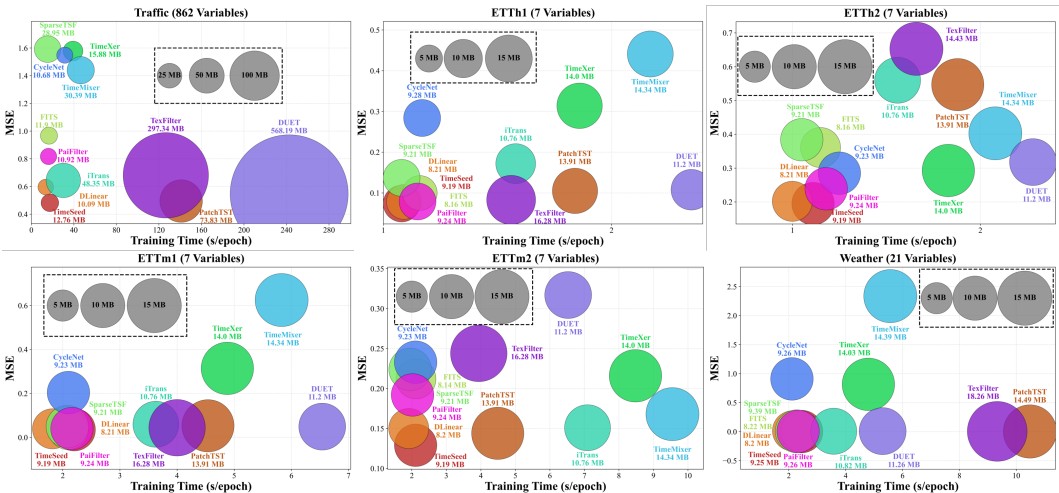

Figure 8: Comparison of model efficiency in the input-96-predict-96 setting.

# E FULL RESULTS OF FREQUENCY-DOMAIN ANALYSIS

To further evaluate the ability of the Frequency-Domain Aggregator (FDA) to reconstruct the trend component of historical endogenous variables, we provide additional prediction cases in Figure 9. The results demonstrate that Frequency-domain Aggregator effectively leverages the trend-related information in exogenous variables and establishes robust correlations between the trends of exogenous variables and those of historical endogenous variables, thereby enabling effective reconstruction of the target trend components.

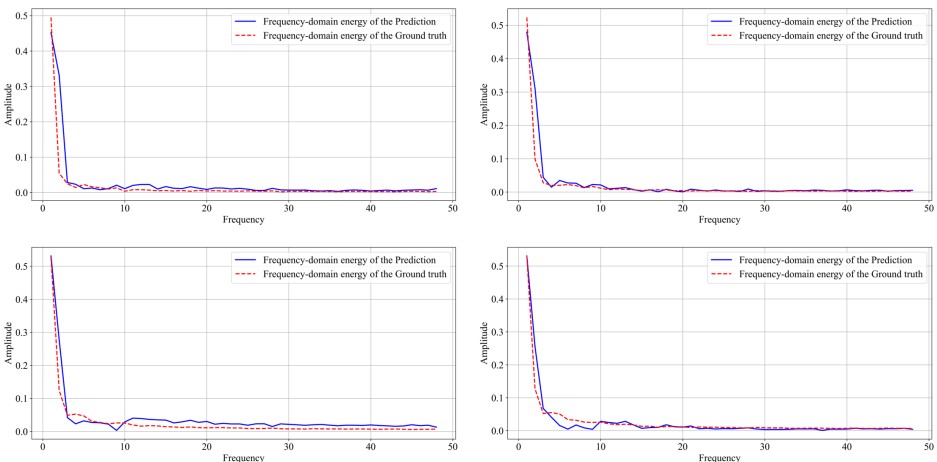

Figure 9: Frequency spectrum of the reconstructed trend component of historical endogenous variables (blue) and the ground truth (red) in the frequency domain. The closer the two curves, the better the reconstruction performance.

# F    FULL RESULTS OF HYPERPARAMETER SENSITIVITY ANALYSIS

In the frequency-domain decomposition module, we further split the input trend component based on an energy-oriented perspective. To explore how different decomposition thresholds affect model performance, we report the complete results of varying $K$ values on the ECL and Traffic datasets in Figures 10, 11, 12 and 13. It is evident that the choice of decomposition threshold, reflected by different values of $K$, has a considerable impact on model performance. Overall, on both the ECL and Traffic datasets, as $K$ increases, the error metrics first decrease and then gradually stabilize, with a slight rise in some cases. This indicates that excessively small thresholds fail to adequately capture the energy characteristics of the input sequence, whereas overly large thresholds may introduce redundant decomposition components, thereby impairing the model's generalization ability. The best performance is generally observed at moderate $K$ values, suggesting that an appropriate threshold strikes a balance between preserving dominant trends and retaining local variations. In summary, the predictive performance across different $K$ values does not vary drastically, further demonstrating the robustness of the TimeSeed.

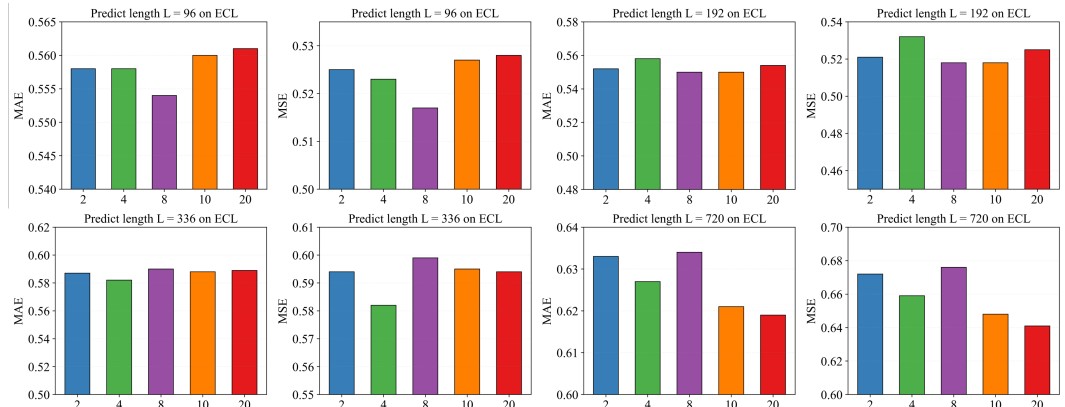

Figure 10: Impact of different $K$ values in ENERGY-DECOMPOSE within the Frequency Domain Aggregator, evaluated on the ECL dataset with various prediction lengths {96, 192, 336, 720} based on a single endogenous setting.

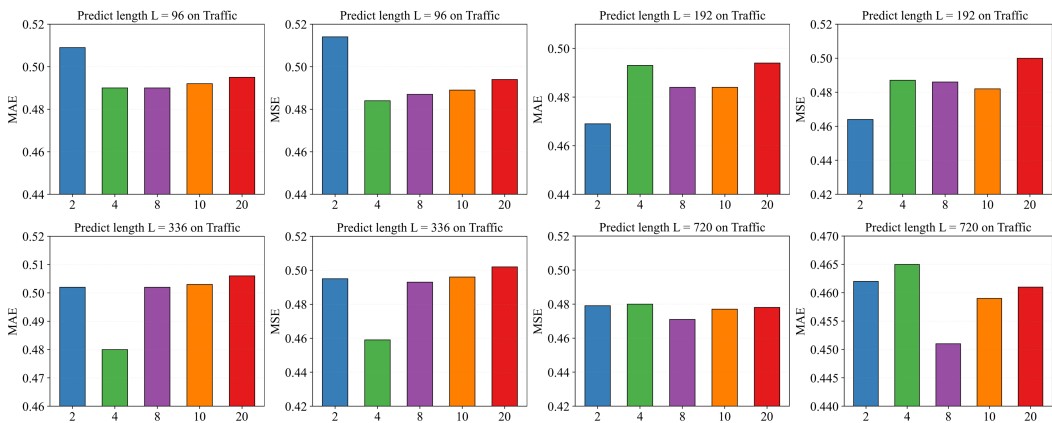

Figure 11: Impact of different $K$ values in ENERGY-DECOMPOSE within the Frequency Domain Aggregator, evaluated on the Traffic dataset with various prediction lengths {96, 192, 336, 720} based on a single endogenous setting.

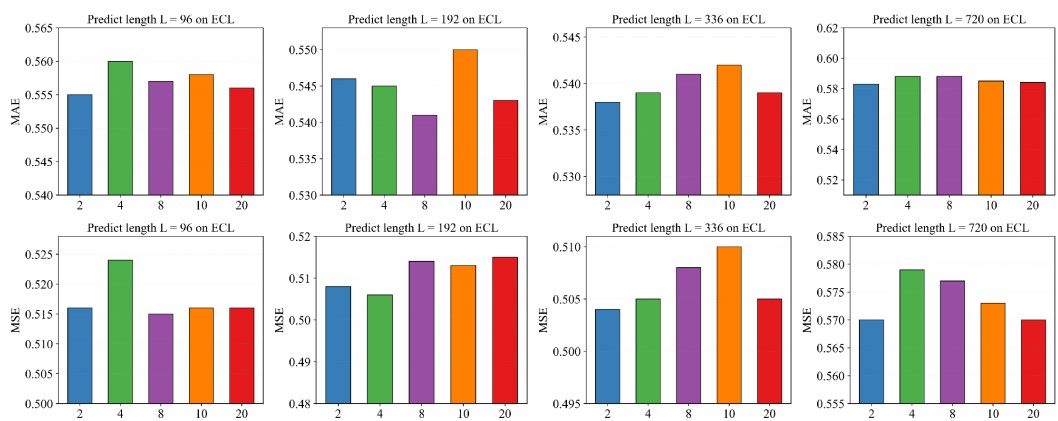

Figure 12: Impact of different $K$ values in Energy-Decompose within the Frequency Domain Aggregator, evaluated on the ECL dataset with various prediction lengths {96, 192, 336, 720} based on sparse endogenous variable setting.

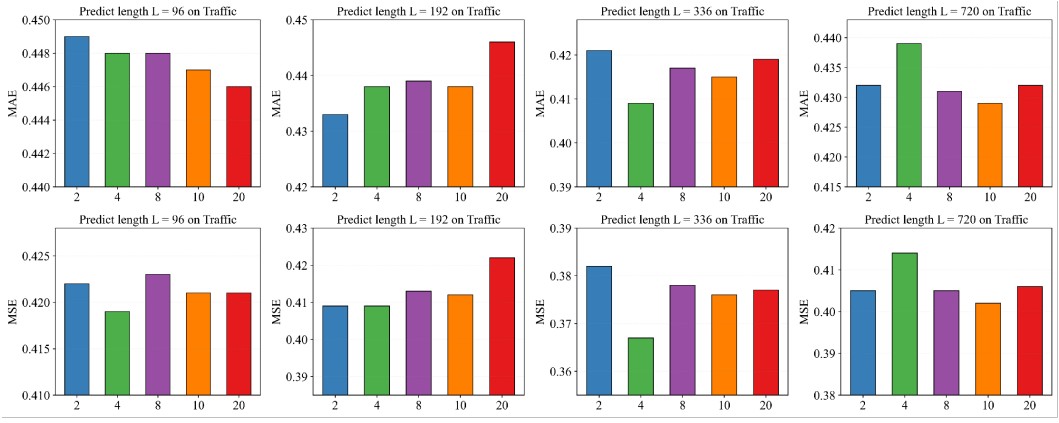

Figure 13: Impact of different $K$ values in Energy-Decompose within the Frequency Domain Aggregator, evaluated on the Traffic dataset with various prediction lengths {96, 192, 336, 720} based on sparse endogenous variable setting.

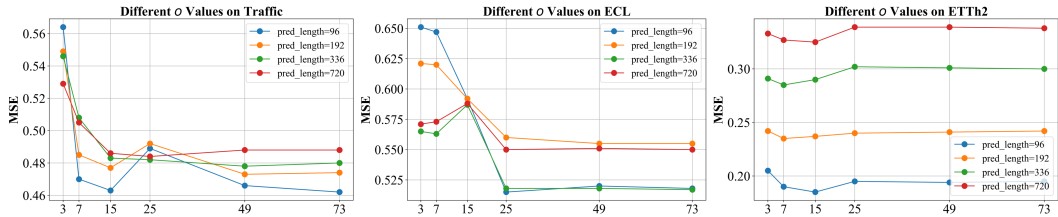

Figure 14: Performance of different downsampling kernels in the Decompose module for long-term time series forecasting on Traffic, ECL and ETTh2 with input-96-predict-96 setting.

We evaluate the sensitivity of TimeSeed across different average pooling kernel sizes ($o$) and high-energy component numbers ($K$). As shown in Figure 14, we investigate the impact of different downsampling kernel sizes ($o$) on the forecasting performance of TimeSeed across various prediction lengths. The results indicate that as the value of $o$ increases, the model performance tends to stabilize or improve. This trend is particularly evident on the Traffic and ECL datasets. In contrast, the ETTh2 dataset exhibits relatively consistent performance regardless of kernel size variations. We also observe that the most effective kernel sizes often align with the data's inherent daily, weekly, or monthly periodicities. Based on these findings, we fix the downsampling kernel size to $o = 25$ for all subsequent experiments.

## G   Full Result of Randomly Sampling Endogenous Variable

Table 10 reports the complete prediction results of TimeSeed, DLinear, TimeXer, and FITS under the strategy based on randomly sampling and uniform interval sampling. Random sampling is closer to real-world forecasting scenarios, and the comparison between the two sampling strategies reflects the impact of different sampling schemes on model performance. The results demonstrate that random sampling consistently yields inferior results relative to uniform interval sampling across most datasets, with this performance gap widening as forecasting horizons extend. This may be attributed to the fact that maintaining uniform temporal structure is more conducive to reconstructing complete historical endogenous sequences, whereas the uncertainty introduced by random sampling can reduce the model's generalization ability. Notably, across both sampling strategies and all forecasting horizons, TimeSeed consistently outperforms the baseline methods, demonstrating its robustness and effectiveness even in challenging sparse scenarios.

Table 10: Full results of unified hyperparameter for long-term time series forecasting results are based on sparse endogenous setting, with the point selection strategy randomly choosing a time step. The look-back window length is fixed at 96. The reported results represent the average performance across different forecasting horizons $L = \{96, 192, 336, 720\}$. $^*$ indicates the use of a random sampling strategy. Lower MSE and MAE values indicate better forecasting performance.

| Model | | TimeSeed | | TimeSeed$^*$ | | DLinear | | DLinear$^*$ | | TimeXer | | TimeXer$^*$ | | FITS | | FITS$^*$ | |
|---|---|---|---|---|---|---|---|---|---|---|---|---|---|---|---|---|---|
| Metric | | MSE | MAE | MSE | MAE | MSE | MAE | MSE | MAE | MSE | MAE | MSE | MAE | MSE | MAE | MSE | MAE |
| ETTh1 | 96 | **0.069** | **0.203** | 0.074 | 0.209 | 0.079 | 0.218 | 0.081 | 0.219 | 0.438 | 0.572 | 0.513 | 0.625 | 0.089 | 0.230 | 0.089 | 0.230 |
| | 192 | **0.091** | **0.236** | 0.100 | 0.241 | 0.105 | 0.251 | 0.106 | 0.251 | 0.442 | 0.565 | 0.413 | 0.540 | 0.211 | 0.372 | 0.211 | 0.372 |
| | 336 | **0.111** | **0.262** | 0.121 | 0.269 | 0.138 | 0.292 | 0.138 | 0.292 | 0.429 | 0.552 | 0.537 | 0.635 | 0.120 | 0.273 | 0.120 | 0.273 |
| | 720 | **0.112** | **0.265** | 0.189 | 0.350 | 0.131 | 0.288 | 0.130 | 0.288 | 0.464 | 0.588 | 0.643 | 0.715 | 0.149 | 0.311 | 0.148 | 0.310 |
| | AVG | **0.096** | **0.242** | 0.121 | 0.267 | 0.113 | 0.262 | 0.114 | 0.263 | 0.444 | 0.569 | 0.527 | 0.629 | 0.142 | 0.297 | 0.142 | 0.296 |
| ETTh2 | 96 | **0.186** | **0.336** | 0.192 | 0.342 | 0.201 | 0.350 | 0.201 | 0.350 | 0.289 | 0.423 | 0.284 | 0.418 | 0.303 | 0.431 | 0.326 | 0.446 |
| | 192 | **0.234** | **0.379** | 0.234 | 0.380 | 0.255 | 0.397 | 0.252 | 0.396 | 0.330 | 0.457 | 0.325 | 0.455 | 0.333 | 0.460 | 0.355 | 0.475 |
| | 336 | **0.288** | **0.426** | 0.289 | 0.427 | 0.306 | 0.440 | 0.307 | 0.441 | 0.358 | 0.479 | 0.356 | 0.478 | 0.496 | 0.571 | 0.507 | 0.578 |
| | 720 | **0.380** | **0.499** | 0.397 | 0.510 | 0.873 | 0.786 | 0.848 | 0.769 | 0.390 | 0.501 | 0.389 | 0.500 | 1.079 | 0.887 | 1.076 | 0.882 |
| | AVG | **0.272** | **0.410** | 0.278 | 0.415 | 0.409 | 0.493 | 0.402 | 0.489 | 0.342 | 0.465 | 0.339 | 0.463 | 0.553 | 0.587 | 0.566 | 0.595 |
| ETTm1 | 96 | **0.036** | **0.143** | 0.038 | 0.148 | 0.043 | 0.157 | 0.047 | 0.166 | 0.290 | 0.468 | 0.292 | 0.472 | 0.051 | 0.175 | 0.051 | 0.175 |
| | 192 | **0.055** | **0.179** | 0.058 | 0.183 | 0.064 | 0.191 | 0.066 | 0.193 | 0.301 | 0.468 | 0.329 | 0.490 | 0.083 | 0.226 | 0.082 | 0.223 |
| | 336 | **0.076** | **0.210** | 0.077 | 0.211 | 0.083 | 0.218 | 0.085 | 0.222 | 0.359 | 0.508 | 0.333 | 0.491 | 0.092 | 0.231 | 0.096 | 0.236 |
| | 720 | 0.106 | 0.246 | 0.105 | 0.244 | 0.111 | 0.253 | 0.115 | 0.258 | 0.426 | 0.551 | 0.451 | 0.563 | **0.088** | 0.232 | 0.088 | **0.229** |
| | AVG | **0.068** | **0.194** | 0.070 | 0.196 | 0.075 | 0.205 | 0.078 | 0.210 | 0.344 | 0.499 | 0.351 | 0.504 | 0.079 | 0.216 | 0.079 | 0.216 |
| ETTm2 | 96 | **0.092** | **0.224** | 0.119 | 0.260 | 0.093 | 0.224 | 0.132 | 0.272 | 0.239 | 0.377 | 0.228 | 0.365 | 0.118 | 0.260 | 0.192 | 0.338 |
| | 192 | **0.127** | **0.269** | 0.151 | 0.297 | 0.135 | 0.276 | 0.164 | 0.309 | 0.245 | 0.384 | 0.259 | 0.395 | 0.151 | 0.298 | 0.201 | 0.346 |
| | 336 | **0.160** | **0.304** | 0.182 | 0.327 | 0.171 | 0.315 | 0.198 | 0.342 | 0.261 | 0.404 | 0.283 | 0.417 | 0.212 | 0.357 | 0.250 | 0.388 |
| | 720 | **0.217** | **0.360** | 0.231 | 0.373 | 0.262 | 0.398 | 0.292 | 0.423 | 0.317 | 0.448 | 0.320 | 0.449 | 0.328 | 0.457 | 0.436 | 0.534 |
| | AVG | **0.149** | **0.289** | 0.170 | 0.314 | 0.165 | 0.303 | 0.196 | 0.337 | 0.266 | 0.404 | 0.273 | 0.406 | 0.202 | 0.343 | 0.269 | 0.402 |

To further evaluate the model's performance under extreme conditions, we increased the prediction difficulty by considering scenarios where the endogenous series is extremely sparse, retaining only a single observation. As shown in Table 11, under this more challenging setting, the overall performance of all models declines. Notably, TimeSeed still achieves the best predictive performance. This can likely be attributed to our multi-stage problem decomposition strategy and reconstruction-based learning mechanism, which effectively enhance TimeSeed's generalization ability, allowing it to maintain high prediction accuracy even when available information is severely limited.

Table 11: Full results of unified hyperparameter for long-term time series forecasting results are based on a single endogenous setting, with the point selection strategy randomly sampling a time step. The look-back window length is fixed at 96. The reported results represent the average performance across different forecasting horizons $L = \{96, 192, 336, 720\}$. $^*$ indicates the use of a random sampling strategy. Lower MSE and MAE values indicate better forecasting performance.

| Model | | TimeSeed$^*$ | | TimeSeed | | DLinear$^*$ | | DLinear | | TimeXer$^*$ | | TimeXer | | FITS$^*$ | | FITS | |
|---|---|---|---|---|---|---|---|---|---|---|---|---|---|---|---|---|---|
| Metric | | MSE | MAE | MSE | MAE | MSE | MAE | MSE | MAE | MSE | MAE | MSE | MAE | MSE | MAE | MSE | MAE |
| ETTh1 | 96 | **0.096** | **0.243** | **0.074** | **0.213** | 0.104 | 0.262 | 0.078 | 0.221 | 0.402 | 0.529 | 0.314 | 0.469 | 0.131 | 0.291 | 0.100 | 0.251 |
| | 192 | **0.113** | **0.266** | **0.098** | **0.244** | 0.132 | 0.292 | 0.104 | 0.253 | 0.405 | 0.543 | 0.340 | 0.491 | 0.140 | 0.301 | 0.115 | 0.268 |
| | 336 | **0.126** | **0.283** | **0.119** | **0.271** | 0.151 | 0.309 | 0.133 | 0.287 | 0.474 | 0.596 | 0.393 | 0.537 | 0.156 | 0.317 | 0.139 | 0.293 |
| | 720 | **0.117** | **0.272** | **0.114** | **0.267** | 0.156 | 0.319 | 0.148 | 0.310 | 0.509 | 0.621 | 0.476 | 0.602 | 0.148 | 0.310 | 0.137 | 0.297 |
| | AVG | **0.113** | **0.266** | **0.101** | **0.249** | 0.136 | 0.295 | 0.116 | 0.268 | 0.448 | 0.572 | 0.380 | 0.525 | 0.144 | 0.305 | 0.123 | 0.278 |
| ETTh2 | 96 | **0.291** | **0.432** | **0.195** | **0.343** | 0.297 | 0.437 | 0.202 | 0.350 | 0.370 | 0.488 | 0.293 | 0.427 | 0.367 | 0.484 | 0.361 | 0.472 |
| | 192 | **0.340** | **0.471** | **0.240** | **0.386** | 0.346 | 0.478 | 0.254 | 0.404 | 0.424 | 0.525 | 0.316 | 0.445 | 0.545 | 0.596 | 0.501 | 0.570 |
| | 336 | **0.378** | **0.500** | **0.302** | **0.439** | 0.393 | 0.509 | 0.308 | 0.444 | 0.425 | 0.526 | 0.358 | 0.479 | 0.788 | 0.732 | 0.603 | 0.635 |
| | 720 | **0.398** | **0.508** | **0.339** | **0.467** | 0.545 | 0.596 | 0.420 | 0.521 | 0.436 | 0.531 | 0.409 | 0.513 | 1.619 | 1.119 | 1.430 | 1.043 |
| | AVG | **0.352** | **0.478** | **0.269** | **0.409** | 0.395 | 0.505 | 0.296 | 0.430 | 0.414 | 0.518 | 0.344 | 0.466 | 0.830 | 0.733 | 0.723 | 0.680 |
| ETTm1 | 96 | **0.054** | **0.184** | **0.035** | **0.143** | 0.054 | 0.184 | 0.039 | 0.151 | 0.331 | 0.500 | 0.314 | 0.492 | 0.054 | 0.183 | 0.035 | 0.144 |
| | 192 | 0.066 | 0.204 | **0.051** | 0.175 | 0.068 | 0.206 | 0.054 | 0.179 | 0.359 | 0.519 | 0.324 | 0.495 | **0.065** | **0.200** | **0.051** | **0.172** |
| | 336 | **0.077** | **0.219** | **0.067** | **0.202** | 0.081 | 0.226 | 0.071 | 0.207 | 0.370 | 0.520 | 0.317 | 0.478 | 0.077 | 0.220 | 0.067 | 0.203 |
| | 720 | **0.097** | **0.246** | **0.091** | **0.235** | 0.106 | 0.261 | 0.098 | 0.248 | 0.347 | 0.496 | 0.344 | 0.491 | 0.102 | 0.257 | 0.092 | 0.242 |
| | AVG | **0.073** | **0.213** | **0.061** | **0.189** | 0.077 | 0.219 | 0.065 | 0.196 | 0.352 | 0.509 | 0.325 | 0.489 | 0.074 | 0.215 | 0.061 | 0.190 |
| ETTm2 | 96 | **0.214** | **0.367** | **0.129** | **0.271** | 0.222 | 0.371 | 0.150 | 0.290 | 0.271 | 0.409 | 0.216 | 0.348 | 0.258 | 0.397 | 0.212 | 0.360 |
| | 192 | **0.237** | **0.385** | **0.159** | **0.304** | 0.247 | 0.392 | 0.179 | 0.325 | 0.288 | 0.422 | 0.248 | 0.387 | 0.274 | 0.411 | 0.242 | 0.376 |
| | 336 | **0.260** | **0.402** | **0.187** | **0.335** | 0.263 | 0.404 | 0.205 | 0.350 | 0.334 | 0.456 | 0.285 | 0.418 | 0.399 | 0.508 | 0.269 | 0.421 |
| | 720 | **0.295** | **0.434** | **0.230** | **0.376** | 0.381 | 0.493 | 0.289 | 0.424 | 0.347 | 0.472 | 0.323 | 0.452 | 0.479 | 0.563 | 0.270 | 0.540 |
| | AVG | **0.251** | **0.397** | **0.176** | **0.321** | 0.278 | 0.415 | 0.206 | 0.347 | 0.310 | 0.440 | 0.268 | 0.401 | 0.353 | 0.470 | 0.248 | 0.424 |

## H  FULL RESULTS OF ABLATION

To validate the effectiveness of the TimeSeed architecture, we conducted comprehensive ablation studies on all modules. The results are presented in Tables 12.

Table 12: Full results of Ablation study results on the key components of TimeSeed. PI denotes Patch-Interact (Eq. (3)), ASR denotes Adaptive Scale Reconstructor, FDA denotes Frequency Domain Aggregator, and TDA denotes Time Domain Aggregator. Moreover,, AggHigh and AggLow are defined in Eq. (7).

| Model | | Ours | | w/o Agghigh | | w/o Agglow | | w/o P-I | | w/o FDA | | w/o TDA | | w/o Sparse point | | Direct Forecasting | |
|---|---|---|---|---|---|---|---|---|---|---|---|---|---|---|---|---|---|
| Metric | | MSE | MAE | MSE | MAE | MSE | MAE | MSE | MAE | MSE | MAE | MSE | MAE | MSE | MAE | MSE | MAE |
| ETTh1 | 96 | **0.069** | **0.203** | 0.078 | 0.203 | 0.075 | 0.213 | 0.072 | 0.206 | 0.073 | 0.207 | 0.073 | 0.204 | 0.071 | 0.209 | 0.083 | 0.223 |
| | 192 | **0.091** | **0.236** | 0.098 | 0.236 | 0.098 | 0.245 | 0.097 | 0.238 | 0.097 | 0.238 | 0.095 | 0.234 | 0.095 | 0.240 | 0.122 | 0.268 |
| | 336 | **0.111** | **0.262** | 0.118 | 0.263 | 0.124 | 0.275 | 0.123 | 0.272 | 0.121 | 0.269 | 0.119 | 0.266 | 0.114 | 0.266 | 0.147 | 0.303 |
| | 720 | **0.112** | **0.265** | 0.153 | 0.265 | 0.142 | 0.295 | 0.183 | 0.344 | 0.182 | 0.344 | 0.183 | 0.345 | 0.117 | 0.271 | 0.143 | 0.302 |
| | AVG | **0.096** | **0.242** | 0.112 | 0.242 | 0.110 | 0.257 | 0.119 | 0.265 | 0.118 | 0.265 | 0.117 | 0.263 | 0.099 | 0.246 | 0.124 | 0.274 |
| ETTm1 | 96 | **0.036** | **0.143** | 0.037 | 0.145 | 0.043 | 0.155 | 0.037 | 0.146 | 0.039 | 0.151 | 0.040 | 0.154 | 0.042 | 0.160 | 0.044 | 0.159 |
| | 192 | **0.055** | **0.179** | 0.057 | 0.181 | 0.058 | 0.182 | 0.057 | 0.181 | 0.059 | 0.184 | 0.058 | 0.184 | 0.058 | 0.186 | 0.064 | 0.190 |
| | 336 | **0.076** | **0.210** | 0.097 | 0.238 | 0.078 | 0.213 | 0.078 | 0.212 | 0.099 | 0.241 | 0.086 | 0.231 | 0.088 | 0.233 | 0.079 | 0.216 |
| | 720 | **0.106** | **0.246** | 0.146 | 0.297 | 0.127 | 0.272 | 0.125 | 0.268 | 0.154 | 0.311 | 0.148 | 0.308 | 0.142 | 0.296 | 0.110 | 0.253 |
| | AVG | **0.068** | **0.194** | 0.084 | 0.215 | 0.077 | 0.206 | 0.074 | 0.202 | 0.088 | 0.222 | 0.083 | 0.219 | 0.083 | 0.219 | 0.074 | 0.204 |
| ETTh2 | 96 | **0.186** | **0.336** | 0.197 | 0.345 | 0.194 | 0.343 | 0.202 | 0.350 | 0.205 | 0.354 | 0.287 | 0.422 | 0.194 | 0.343 | 0.217 | 0.364 |
| | 192 | **0.234** | **0.379** | 0.237 | 0.380 | 0.260 | 0.397 | 0.259 | 0.399 | 0.270 | 0.399 | 0.336 | 0.461 | 0.239 | 0.386 | 0.287 | 0.423 |
| | 336 | **0.288** | **0.426** | 0.299 | 0.438 | 0.297 | 0.434 | 0.296 | 0.433 | 0.304 | 0.441 | 0.390 | 0.501 | 0.294 | 0.433 | 0.341 | 0.465 |
| | 720 | **0.380** | **0.499** | 0.403 | 0.515 | 0.388 | 0.505 | 0.406 | 0.516 | 0.417 | 0.522 | 0.488 | 0.566 | 0.388 | 0.496 | 0.754 | 0.719 |
| | AVG | **0.272** | **0.410** | 0.284 | 0.420 | 0.285 | 0.420 | 0.291 | 0.424 | 0.299 | 0.432 | 0.376 | 0.488 | 0.279 | 0.415 | 0.400 | 0.493 |
| ETTm2 | 96 | **0.092** | **0.224** | 0.097 | 0.229 | 0.096 | 0.229 | 0.099 | 0.234 | 0.093 | 0.225 | 0.100 | 0.233 | 0.129 | 0.272 | 0.096 | 0.228 |
| | 192 | **0.127** | **0.269** | 0.132 | 0.275 | 0.131 | 0.273 | 0.135 | 0.279 | 0.132 | 0.275 | 0.132 | 0.274 | 0.158 | 0.304 | 0.131 | 0.273 |
| | 336 | **0.160** | **0.304** | 0.165 | 0.310 | 0.179 | 0.323 | 0.168 | 0.313 | 0.166 | 0.315 | 0.167 | 0.312 | 0.186 | 0.333 | 0.186 | 0.329 |
| | 720 | **0.217** | **0.360** | 0.224 | 0.367 | 0.229 | 0.371 | 0.220 | 0.363 | 0.223 | 0.357 | 0.219 | 0.362 | 0.231 | 0.377 | 0.228 | 0.371 |
| | AVG | **0.149** | **0.289** | 0.155 | 0.295 | 0.159 | 0.299 | 0.156 | 0.298 | 0.154 | 0.293 | 0.154 | 0.295 | 0.176 | 0.322 | 0.160 | 0.300 |

## I  FULL RESULTS OF HARD AND SOFT CHOOSE

The results are presented in Tables 13.

Table 13: Full Results of Hard and Soft Choose

| Model | | TimeSeed | | TimeSeed(Soft) | |
|---|---|---|---|---|---|
| Metric | | MSE | MAE | MSE | MAE |
| ETTh1 | 96 | **0.069** | **0.203** | 0.075 | 0.211 |
| | 192 | **0.091** | **0.236** | 0.097 | 0.238 |
| | 336 | **0.111** | **0.262** | 0.121 | 0.269 |
| | 720 | **0.112** | **0.265** | 0.185 | 0.346 |
| | AVG | **0.096** | **0.242** | 0.119 | 0.266 |
| ETTm1 | 96 | **0.036** | **0.143** | 0.038 | 0.147 |
| | 192 | **0.055** | **0.179** | 0.087 | 0.225 |
| | 336 | **0.076** | **0.210** | 0.077 | 0.211 |
| | 720 | **0.106** | **0.246** | 0.168 | 0.323 |
| | AVG | **0.068** | **0.194** | 0.092 | 0.226 |
| Traffic | 96 | **0.421** | **0.447** | 0.547 | 0.572 |
| | 192 | **0.412** | **0.438** | 0.508 | 0.547 |
| | 336 | **0.376** | **0.417** | 0.561 | 0.572 |
| | 720 | **0.402** | **0.429** | 0.563 | 0.579 |
| | AVG | **0.403** | **0.433** | 0.545 | 0.567 |
| Weather | 96 | **0.002** | **0.028** | 0.007 | 0.063 |
| | 192 | **0.002** | **0.033** | 0.003 | 0.042 |
| | 336 | **0.002** | **0.035** | 0.014 | 0.092 |
| | 720 | **0.003** | **0.040** | 0.013 | 0.088 |
| | AVG | **0.002** | **0.034** | 0.009 | 0.071 |

## J    FULL RESULTS WITH EXOGENOUS VARIABLES ONLY

To further investigate the reconstruction capability of TimeSeed for endogenous variables, we report in Table 14 the complete prediction results obtained using only exogenous variables, without endogenous inputs. It is worth noting that a small number of models become distorted under these conditions, with prediction performance completely deteriorating, such as TimeMixer's performance on Weather. In contrast, TimeSeed models the trend and cyclical components of historical endogenous variables through decomposition, thereby enhancing robustness. Finally, it uses reconstructed historical endogenous variables to predict future changes in endogenous variables. This complex prediction problem is decomposed into several simpler subproblems for solution. As a result, it achieves satisfactory performance even under this challenging prediction setting.

Table 14: Comparison of model performance using exogenous variables only.

| Model | | TimeSeed | | iTrans | | TimeXer | | TimeMixer | |
|---|---|---|---|---|---|---|---|---|---|
| Metric | | MSE | MAE | MSE | MAE | MSE | MAE | MSE | MAE |
| Traffic | 96 | 0.408 | 0.459 | **0.262** | **0.332** | 0.595 | 0.659 | 0.502 | 0.593 |
| | 192 | **0.434** | **0.453** | 0.798 | 0.740 | 0.601 | 0.662 | 0.581 | 0.648 |
| | 336 | 0.436 | 0.455 | **0.386** | **0.414** | 0.622 | 0.672 | 0.611 | 0.663 |
| | 720 | **0.417** | **0.447** | 1.031 | 0.860 | 0.713 | 0.726 | 0.671 | 0.705 |
| | AVG | **0.424** | **0.453** | 0.619 | 0.587 | 0.633 | 0.680 | 0.591 | 0.652 |
| Weather | 96 | **0.007** | **0.069** | 0.011 | 0.091 | 0.885 | 0.809 | 2.200 | 1.291 |
| | 192 | 0.007 | 0.069 | **0.006** | **0.062** | 0.754 | 0.732 | 2.363 | 1.370 |
| | 336 | **0.007** | **0.068** | 0.008 | 0.077 | 0.810 | 0.765 | 2.216 | 1.330 |
| | 720 | **0.007** | **0.071** | 0.010 | 0.076 | 0.757 | 0.742 | 2.127 | 1.277 |
| | AVG | **0.007** | **0.069** | 0.009 | 0.076 | 0.801 | 0.762 | 2.227 | 1.317 |

## K    ERROR BARS

Here, we repeat all the experiments five times and report the standard deviation and the statistical significance test in Table 15.

Table 15: Standard deviation and statistical tests for our method.

| Model | TimeSeed | | Confidence |
|---|---|---|---|
| Dataset | MSE | MSE | Interval |
| Weather | 0.002 ± 0.002 | 0.035 ± 0.007 | 0.99 |
| ECL | 0.533 ± 0.035 | 0.560 ± 0.019 | 0.99 |
| Traffic | 0.404 ± 0.017 | 0.435 ± 0.014 | 0.99 |
| ETTh1 | 0.097 ± 0.003 | 0.243 ± 0.005 | 0.99 |
| ETTh2 | 0.271 ± 0.004 | 0.409 ± 0.003 | 0.99 |
| ETTm1 | 0.068 ± 0.003 | 0.194 ± 0.004 | 0.99 |
| ETTm2 | 0.148 ± 0.002 | 0.289 ± 0.001 | 0.99 |

## L    FULL RESULTS OF NON-OVERLAP VS OVERLAP PATCH

In order to better establish the mapping between the periodic components of exogenous and endogenous variables, TimeSeed employs the PATCH mechanism in the Time Domain Aggregator (TDA) to more efficiently fit the periodicity that inherently exists in the dataset. Here, we report the complete results for both non-overlapping (TimeSeed) and overlapping patches (TimeSeed-OL). Based on the patch parameter settings in (Nie et al., 2022), we adopt a step size of 12 and a patch length of 24. As shown in Tables 16 and 17, the non-overlapping patch strategy (TimeSeed) and the overlapping patch strategy (TimeSeed-OL) exhibit substantial performance differences across datasets. The most pronounced disparity is observed on the ETTh1 dataset, where the performance gap reaches 61.6% in the 720-step forecasting task. Furthermore, TimeSeed consistently achieves superior average performance on ETTh1, which may be attributed to its ability to more effectively capture independent temporal features while mitigating the information redundancy introduced by overlapping regions. This conclusion is consistent with the experimental results in (Wang et al., 2024b).

Table 16: Comprehensive performance results of models using overlapping patches based on sparse endogenous variable setting (OL indicates the use of overlapping patches).

| Model | | TimeSeed | | TimeSeed-OL | |
|---|---|---|---|---|---|
| Metric | | MSE | MAE | MSE | MAE |
| ETTh1 | 96 | 0.069 | 0.203 | 0.075 | 0.211 |
| | 192 | 0.091 | 0.236 | 0.094 | 0.236 |
| | 336 | 0.111 | 0.262 | 0.122 | 0.271 |
| | 720 | 0.112 | 0.265 | 0.181 | 0.342 |
| | AVG | 0.096 | 0.242 | 0.118 | 0.265 |
| ETTh2 | 96 | 0.186 | 0.336 | 0.191 | 0.341 |
| | 192 | 0.234 | 0.379 | 0.232 | 0.378 |
| | 336 | 0.288 | 0.426 | 0.287 | 0.425 |
| | 720 | 0.380 | 0.499 | 0.386 | 0.504 |
| | AVG | 0.272 | 0.410 | 0.274 | 0.412 |
| ETTm1 | 96 | 0.036 | 0.143 | 0.037 | 0.146 |
| | 192 | 0.055 | 0.179 | 0.06 | 0.185 |
| | 336 | 0.076 | 0.210 | 0.077 | 0.211 |
| | 720 | 0.106 | 0.246 | 0.105 | 0.244 |
| | AVG | 0.068 | 0.194 | 0.07 | 0.197 |
| ETTm2 | 96 | 0.092 | 0.224 | 0.094 | 0.226 |
| | 192 | 0.127 | 0.269 | 0.128 | 0.27 |
| | 336 | 0.160 | 0.304 | 0.161 | 0.306 |
| | 720 | 0.217 | 0.360 | 0.213 | 0.356 |
| | AVG | 0.149 | 0.289 | 0.149 | 0.29 |
| Traffic | 96 | 0.421 | 0.447 | 0.422 | 0.446 |
| | 192 | 0.412 | 0.438 | 0.423 | 0.447 |
| | 336 | 0.376 | 0.417 | 0.407 | 0.44 |
| | 720 | 0.402 | 0.429 | 0.383 | 0.417 |
| | AVG | 0.403 | 0.433 | 0.409 | 0.437 |
| ECL | 96 | 0.516 | 0.555 | 0.512 | 0.554 |
| | 192 | 0.514 | 0.550 | 0.508 | 0.546 |
| | 336 | 0.510 | 0.542 | 0.539 | 0.563 |
| | 720 | 0.576 | 0.586 | 0.572 | 0.584 |
| | AVG | 0.529 | 0.558 | 0.532 | 0.562 |

Table 17: Comprehensive performance results of models using overlapping patches based on a single endogenous variable setting (OL indicates the use of overlapping patches).

| Model | | TimeSeed | | TimeSeed-OL | |
|---|---|---|---|---|---|
| Metric | | MSE | MAE | MSE | MAE |
| ETTh1 | 96 | 0.074 | 0.213 | 0.070 | 0.209 |
| | 192 | 0.098 | 0.244 | 0.093 | 0.238 |
| | 336 | 0.119 | 0.271 | 0.117 | 0.270 |
| | 720 | 0.114 | 0.267 | 0.114 | 0.267 |
| | AVG | 0.101 | 0.249 | 0.099 | 0.246 |
| ETTh2 | 96 | 0.195 | 0.343 | 0.190 | 0.340 |
| | 192 | 0.240 | 0.386 | 0.238 | 0.386 |
| | 336 | 0.302 | 0.439 | 0.299 | 0.437 |
| | 720 | 0.339 | 0.467 | 0.328 | 0.460 |
| | AVG | 0.269 | 0.409 | 0.264 | 0.406 |
| ETTm1 | 96 | 0.035 | 0.143 | 0.036 | 0.146 |
| | 192 | 0.051 | 0.175 | 0.053 | 0.179 |
| | 336 | 0.067 | 0.202 | 0.068 | 0.204 |
| | 720 | 0.091 | 0.235 | 0.092 | 0.236 |
| | AVG | 0.061 | 0.189 | 0.062 | 0.191 |
| ETTm2 | 96 | 0.129 | 0.271 | 0.129 | 0.271 |
| | 192 | 0.159 | 0.304 | 0.159 | 0.305 |
| | 336 | 0.187 | 0.335 | 0.188 | 0.334 |
| | 720 | 0.230 | 0.376 | 0.230 | 0.375 |
| | AVG | 0.176 | 0.321 | 0.177 | 0.322 |
| ECL | 96 | 0.515 | 0.560 | 0.511 | 0.550 |
| | 192 | 0.518 | 0.550 | 0.611 | 0.602 |
| | 336 | 0.595 | 0.588 | 0.621 | 0.606 |
| | 720 | 0.648 | 0.621 | 0.621 | 0.611 |
| | AVG | 0.569 | 0.580 | 0.591 | 0.592 |
| Traffic | 96 | 0.489 | 0.492 | 0.464 | 0.481 |
| | 192 | 0.482 | 0.484 | 0.477 | 0.489 |
| | 336 | 0.496 | 0.503 | 0.489 | 0.493 |
| | 720 | 0.459 | 0.477 | 0.490 | 0.495 |
| | AVG | 0.482 | 0.489 | 0.480 | 0.489 |

# M    CASE STUDY

To further evaluate our proposed model, we report complementary predictive visualization results on the ETT dataset. For baseline selection, we select representative models, including FITS (Xu et al., 2023), PatchTST (Nie et al., 2022), and DLinear (Zeng et al., 2023). As shown in Figures 15 and 16, TimeSeed performs best in fitting the ground truth, demonstrating excellent predictive performance.

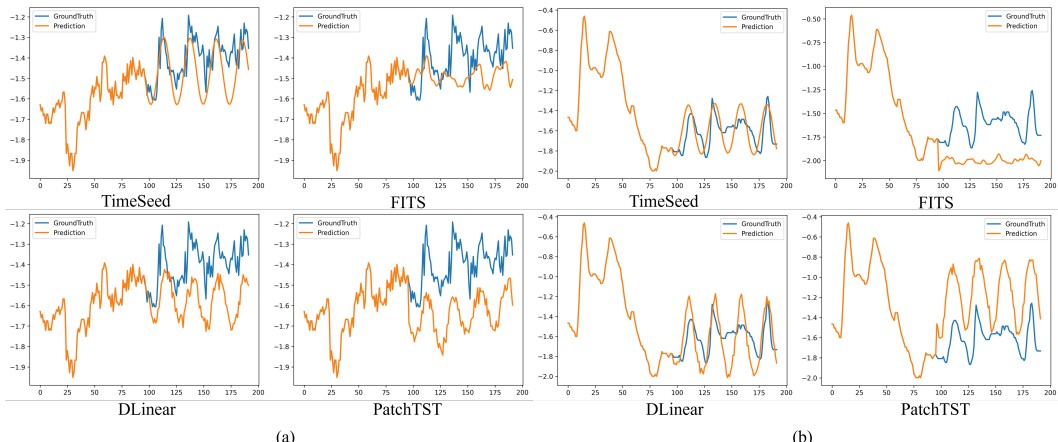

Figure 15: Prediction cases from TimeSeed, FITS, DLinear, and PatchTST on the ETTh1 (a) and ETTh2 (b) datasets under the input-96-predict-96 setting. Blue lines are the ground truths and orange lines are the model predictions.

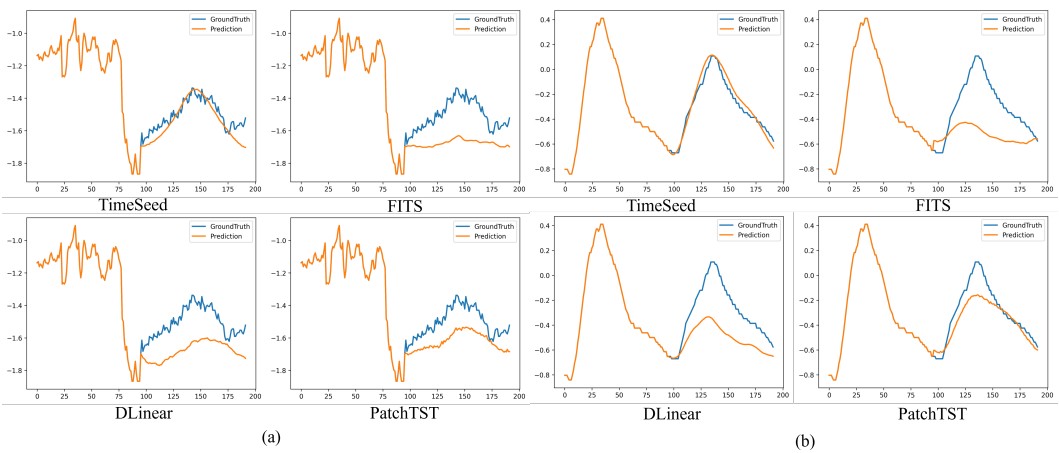

Figure 16: Prediction cases from TimeSeed, FITS, DLinear, and PatchTST on the ETTm1 (a) and ETTm2 (b) datasets under the input-96-predict-96 setting. Blue lines are the ground truths and orange lines are the model predictions.

# N  FULL MAIN RESULTS

Here, we report the results of long time series prediction based on sparse endogenous settings (with missing observations) and one single endogenous settings (without exogenous variables), with prediction lengths including {96, 192, 336, 720}, where the size of the look back window is fixed to 96. All experiments are performed with the unified hyperparameter settings as described earlier. As shown in Tables 18 and 19, TimeSeed achieves optimal performance on almost all metrics compared with baseline models. Notably, under the sparse and single endogenous settings, TimeSeed attains the highest ranking across 59 and 62 evaluation cases ($1^{st}$ Count), respectively, which is substantially more frequent than any baseline model.

Table 18: Full results of unified hyperparameter for long-term time series forecasting results are based on sparse endogenous setting, with the point selection strategy choosing the most recent time step. The look-back window length is fixed at 96. Lower MSE and MAE values indicate better forecasting performance. The best results are highlighted in red, and the second-best results are highlighted in blue.

| Model | | TimeSeed | | DUET | | iTrans | | DLinear | | TimeXer | | TimeMixer | | PatchTST | | FITS | | CycleNet | | TexFilter | | PaiFilter | | SparseTSF | |
|---|---|---|---|---|---|---|---|---|---|---|---|---|---|---|---|---|---|---|---|---|---|---|---|---|---|
| Metric | | MSE | MAE | MSE | MAE | MSE | MAE | MSE | MAE | MSE | MAE | MSE | MAE | MSE | MAE | MSE | MAE | MSE | MAE | MSE | MAE | MSE | MAE | MSE | MAE |
| ETTh1 | 96 | 0.069 | 0.203 | 0.124 | 0.276 | 0.170 | 0.334 | 0.079 | 0.218 | 0.438 | 0.572 | 0.102 | 0.243 | 0.106 | 0.259 | 0.089 | 0.230 | 0.078 | 0.212 | 0.100 | 0.245 | 0.079 | 0.222 | 0.123 | 0.277 |
| | 192 | 0.091 | 0.236 | 0.123 | 0.268 | 0.157 | 0.316 | 0.105 | 0.251 | 0.442 | 0.565 | 0.109 | 0.253 | 0.320 | 0.467 | 0.211 | 0.372 | 0.097 | 0.237 | 0.108 | 0.254 | 0.155 | 0.309 | 0.156 | 0.315 |
| | 336 | 0.111 | 0.262 | 0.180 | 0.336 | 0.248 | 0.403 | 0.138 | 0.292 | 0.429 | 0.552 | 0.267 | 0.413 | 0.205 | 0.356 | 0.120 | 0.273 | 0.109 | 0.255 | 0.133 | 0.287 | 0.242 | 0.402 | 0.178 | 0.338 |
| | 720 | 0.112 | 0.265 | 0.210 | 0.364 | 0.168 | 0.326 | 0.131 | 0.288 | 0.464 | 0.588 | 0.170 | 0.328 | 0.281 | 0.442 | 0.149 | 0.311 | 0.180 | 0.340 | 0.235 | 0.389 | 0.133 | 0.290 | 0.175 | 0.334 |
| | AVG | 0.096 | 0.242 | 0.159 | 0.311 | 0.186 | 0.345 | 0.113 | 0.262 | 0.444 | 0.569 | 0.162 | 0.309 | 0.228 | 0.381 | 0.142 | 0.297 | 0.116 | 0.261 | 0.144 | 0.294 | 0.152 | 0.306 | 0.158 | 0.316 |
| ETTh2 | 96 | 0.186 | 0.336 | 0.363 | 0.475 | 0.581 | 0.655 | 0.201 | 0.350 | 0.289 | 0.423 | 0.213 | 0.355 | 0.436 | 0.535 | 0.303 | 0.431 | 0.198 | 0.345 | 0.368 | 0.482 | 0.251 | 0.395 | 0.377 | 0.490 |
| | 192 | 0.234 | 0.379 | 0.334 | 0.461 | 1.071 | 0.890 | 0.255 | 0.397 | 0.330 | 0.457 | 0.255 | 0.393 | 1.017 | 0.887 | 0.333 | 0.460 | 0.243 | 0.387 | 0.314 | 0.446 | 0.321 | 0.453 | 0.477 | 0.555 |
| | 336 | 0.288 | 0.426 | 0.423 | 0.512 | 0.922 | 0.825 | 0.306 | 0.440 | 0.358 | 0.479 | 0.299 | 0.438 | 0.458 | 0.552 | 0.496 | 0.571 | 0.309 | 0.441 | 0.283 | 0.423 | 0.413 | 0.517 | 0.607 | 0.637 |
| | 720 | 0.380 | 0.499 | 0.406 | 0.517 | 1.034 | 0.873 | 0.873 | 0.786 | 0.390 | 0.501 | 0.331 | 0.462 | 0.384 | 0.494 | 1.079 | 0.887 | 0.414 | 0.519 | 0.347 | 0.473 | 0.915 | 0.815 | 1.028 | 0.859 |
| | AVG | 0.272 | 0.410 | 0.381 | 0.491 | 0.902 | 0.811 | 0.409 | 0.493 | 0.342 | 0.465 | 0.274 | 0.412 | 0.574 | 0.617 | 0.553 | 0.587 | 0.291 | 0.423 | 0.328 | 0.456 | 0.475 | 0.545 | 0.622 | 0.635 |
| ETTm1 | 96 | 0.036 | 0.143 | 0.068 | 0.202 | 0.046 | 0.164 | 0.043 | 0.157 | 0.290 | 0.468 | 0.107 | 0.274 | 0.046 | 0.166 | 0.051 | 0.175 | 0.044 | 0.158 | 0.114 | 0.266 | 0.050 | 0.172 | 0.046 | 0.164 |
| | 192 | 0.055 | 0.179 | 0.094 | 0.237 | 0.074 | 0.209 | 0.064 | 0.191 | 0.301 | 0.468 | 0.137 | 0.306 | 0.068 | 0.196 | 0.083 | 0.226 | 0.094 | 0.241 | 0.055 | 0.180 | 0.069 | 0.205 | 0.065 | 0.196 |
| | 336 | 0.076 | 0.210 | 0.200 | 0.363 | 0.357 | 0.524 | 0.083 | 0.218 | 0.359 | 0.508 | 0.152 | 0.306 | 0.123 | 0.275 | 0.092 | 0.231 | 0.126 | 0.280 | 0.225 | 0.388 | 0.194 | 0.362 | 0.086 | 0.227 |
| | 720 | 0.106 | 0.246 | 0.265 | 0.409 | 0.095 | 0.232 | 0.111 | 0.253 | 0.426 | 0.551 | 0.170 | 0.336 | 0.181 | 0.341 | 0.088 | 0.232 | 0.153 | 0.306 | 0.186 | 0.343 | 0.123 | 0.279 | 1.800 | 1.277 |
| | AVG | 0.068 | 0.194 | 0.157 | 0.303 | 0.143 | 0.282 | 0.075 | 0.205 | 0.344 | 0.499 | 0.141 | 0.305 | 0.104 | 0.245 | 0.079 | 0.216 | 0.104 | 0.246 | 0.145 | 0.294 | 0.109 | 0.254 | 0.499 | 0.466 |
| ETTm2 | 96 | 0.092 | 0.224 | 0.247 | 0.392 | 0.140 | 0.294 | 0.093 | 0.224 | 0.239 | 0.377 | 0.103 | 0.248 | 0.197 | 0.368 | 0.118 | 0.260 | 0.096 | 0.226 | 0.119 | 0.259 | 0.106 | 0.243 | 0.127 | 0.273 |
| | 192 | 0.127 | 0.269 | 0.225 | 0.365 | 0.399 | 0.543 | 0.135 | 0.276 | 0.245 | 0.384 | 0.177 | 0.330 | 0.225 | 0.384 | 0.151 | 0.298 | 0.133 | 0.273 | 0.151 | 0.297 | 0.168 | 0.317 | 0.164 | 0.315 |
| | 336 | 0.160 | 0.304 | 0.313 | 0.443 | 0.381 | 0.517 | 0.171 | 0.315 | 0.261 | 0.404 | 0.160 | 0.310 | 0.188 | 0.333 | 0.212 | 0.357 | 0.171 | 0.315 | 0.178 | 0.324 | 0.200 | 0.345 | 0.201 | 0.350 |
| | 720 | 0.217 | 0.360 | 0.350 | 0.471 | 0.573 | 0.627 | 0.262 | 0.398 | 0.317 | 0.448 | 0.276 | 0.411 | 0.224 | 0.365 | 0.328 | 0.457 | 0.230 | 0.371 | 0.257 | 0.394 | 0.231 | 0.376 | 0.279 | 0.418 |
| | AVG | 0.149 | 0.289 | 0.284 | 0.417 | 0.373 | 0.495 | 0.165 | 0.303 | 0.266 | 0.404 | 0.179 | 0.325 | 0.208 | 0.362 | 0.202 | 0.343 | 0.157 | 0.296 | 0.176 | 0.318 | 0.176 | 0.320 | 0.193 | 0.339 |
| Weather | 96 | 0.002 | 0.028 | 0.007 | 0.059 | 0.009 | 0.078 | 0.014 | 0.081 | 0.758 | 0.751 | 0.005 | 0.063 | 0.003 | 0.041 | 0.005 | 0.060 | 0.004 | 0.043 | 0.003 | 0.038 | 0.005 | 0.050 | 0.005 | 0.059 |
| | 192 | 0.002 | 0.033 | 0.005 | 0.054 | 0.020 | 0.109 | 0.003 | 0.045 | 0.831 | 0.793 | 0.468 | 0.680 | 0.006 | 0.059 | 0.009 | 0.072 | 0.003 | 0.039 | 0.007 | 0.061 | 0.008 | 0.070 | 0.008 | 0.077 |
| | 336 | 0.002 | 0.035 | 0.005 | 0.060 | 0.005 | 0.058 | 0.008 | 0.066 | 0.814 | 0.782 | 0.409 | 0.636 | 0.005 | 0.054 | 0.008 | 0.072 | 0.002 | 0.035 | 0.004 | 0.049 | 0.005 | 0.053 | 0.008 | 0.073 |
| | 720 | 0.003 | 0.040 | 0.006 | 0.058 | 0.008 | 0.076 | 0.009 | 0.074 | 0.905 | 0.831 | 0.692 | 0.828 | 0.004 | 0.046 | 0.006 | 0.062 | 0.003 | 0.040 | 0.006 | 0.060 | 0.030 | 0.134 | 0.006 | 0.064 |
| | AVG | 0.002 | 0.034 | 0.006 | 0.058 | 0.011 | 0.080 | 0.009 | 0.066 | 0.827 | 0.789 | 0.394 | 0.552 | 0.004 | 0.050 | 0.007 | 0.067 | 0.003 | 0.039 | 0.005 | 0.052 | 0.012 | 0.077 | 0.007 | 0.068 |
| ECL | 96 | 0.516 | 0.555 | 0.605 | 0.606 | 1.047 | 0.820 | 0.800 | 0.684 | 0.558 | 0.563 | 1.869 | 1.060 | 0.776 | 0.683 | 1.499 | 1.007 | 0.784 | 0.675 | 0.811 | 0.700 | 0.983 | 0.787 | 0.712 | 0.655 |
| | 192 | 0.514 | 0.550 | 0.579 | 0.582 | 0.693 | 0.666 | 0.824 | 0.695 | 0.533 | 0.549 | 2.369 | 1.239 | 0.624 | 0.620 | 1.329 | 0.912 | 0.780 | 0.669 | 0.724 | 0.640 | 1.043 | 0.823 | 0.764 | 0.684 |
| | 336 | 0.510 | 0.542 | 0.665 | 0.629 | 0.603 | 0.595 | 0.710 | 0.642 | 0.571 | 0.569 | 1.978 | 1.120 | 0.605 | 0.592 | 0.696 | 0.640 | 0.850 | 0.701 | 0.911 | 0.737 | 0.817 | 0.698 | 0.641 | 0.624 |
| | 720 | 0.576 | 0.586 | 0.713 | 0.654 | 0.652 | 0.624 | 0.994 | 0.776 | 0.638 | 0.611 | 0.480 | 0.526 | 0.851 | 0.741 | 0.822 | 0.707 | 0.872 | 0.715 | 0.655 | 0.630 | 1.267 | 0.927 | 0.717 | 0.662 |
| | AVG | 0.529 | 0.558 | 0.641 | 0.618 | 0.749 | 0.676 | 0.832 | 0.699 | 0.575 | 0.573 | 1.674 | 0.987 | 0.714 | 0.659 | 1.087 | 0.817 | 0.822 | 0.690 | 0.775 | 0.677 | 1.027 | 0.809 | 0.709 | 0.656 |
| Traffic | 96 | 0.421 | 0.447 | 0.668 | 0.621 | 0.441 | 0.457 | 0.569 | 0.537 | 1.345 | 1.012 | 0.635 | 0.603 | 0.531 | 0.527 | 0.629 | 0.595 | 0.885 | 0.684 | 0.583 | 0.552 | 0.473 | 0.475 | 1.236 | 0.876 |
| | 192 | 0.412 | 0.438 | 0.543 | 0.538 | 0.646 | 0.607 | 0.485 | 0.481 | 1.436 | 1.056 | 1.363 | 0.874 | 0.516 | 0.516 | 0.653 | 0.609 | 0.875 | 0.676 | 0.683 | 0.597 | 0.946 | 0.764 | 1.220 | 0.859 |
| | 336 | 0.376 | 0.417 | 0.571 | 0.570 | 0.425 | 0.458 | 0.671 | 0.590 | 1.363 | 0.995 | 1.627 | 0.951 | 0.431 | 0.474 | 0.909 | 0.722 | 0.875 | 0.678 | 0.476 | 0.491 | 0.461 | 0.485 | 1.287 | 0.896 |
| | 720 | 0.402 | 0.429 | 0.515 | 0.515 | 0.419 | 0.452 | 0.611 | 0.558 | 1.441 | 1.047 | 1.379 | 0.885 | 0.518 | 0.519 | 0.970 | 0.764 | 0.869 | 0.672 | 0.594 | 0.554 | 0.932 | 0.750 | 1.361 | 0.913 |
| | AVG | 0.403 | 0.433 | 0.574 | 0.561 | 0.483 | 0.493 | 0.584 | 0.542 | 1.396 | 1.028 | 1.251 | 0.828 | 0.499 | 0.509 | 0.790 | 0.672 | 0.876 | 0.678 | 0.584 | 0.548 | 0.703 | 0.619 | 1.276 | 0.886 |
| $1^{st}$ Count | | 30 | 29 | 0 | 0 | 0 | 1 | 0 | 0 | 0 | 1 | 2 | 2 | 0 | 0 | 1 | 0 | 1 | 1 | 1 | 1 | 0 | 0 | 0 | 0 |

Table 19: Full results of unified hyperparameter for long-term time series forecasting results are based on a single endogenous setting, with the point selection strategy choosing the most recent time step. The look-back window length is fixed at 96. Lower MSE and MAE values indicate better forecasting performance. The best results are highlighted in red, and the second-best results are highlighted in blue.

| Model Metric | | TimeSeed | | DUET | | iTrans | | DLinear | | TimeXer | | TimeMixer | | PatchTST | | FITS | | CycleNet | | TexFilter | | PaiFilter | | SparseTSF | |
|---|---|---|---|---|---|---|---|---|---|---|---|---|---|---|---|---|---|---|---|---|---|---|---|---|---|
| | | MSE | MAE | MSE | MAE | MSE | MAE | MSE | MAE | MSE | MAE | MSE | MAE | MSE | MAE | MSE | MAE | MSE | MAE | MSE | MAE | MSE | MAE | MSE | MAE |
| ETTh1 | 96 | 0.074 | 0.213 | 0.108 | 0.259 | 0.172 | 0.332 | 0.078 | 0.221 | 0.314 | 0.469 | 0.441 | 0.582 | 0.105 | 0.257 | 0.100 | 0.251 | 0.284 | 0.444 | 0.083 | 0.230 | 0.079 | 0.221 | 0.137 | 0.295 |
| | 192 | 0.098 | 0.244 | 0.124 | 0.276 | 0.151 | 0.308 | 0.104 | 0.253 | 0.340 | 0.491 | 0.470 | 0.579 | 0.201 | 0.365 | 0.115 | 0.268 | 0.327 | 0.475 | 0.110 | 0.259 | 0.142 | 0.297 | 0.165 | 0.328 |
| | 336 | 0.119 | 0.271 | 0.169 | 0.323 | 0.203 | 0.348 | 0.133 | 0.287 | 0.393 | 0.537 | 0.793 | 0.789 | 0.159 | 0.304 | 0.139 | 0.293 | 0.332 | 0.485 | 0.133 | 0.288 | 0.132 | 0.284 | 0.188 | 0.348 |
| | 720 | 0.114 | 0.267 | 0.357 | 0.472 | 0.189 | 0.342 | 0.148 | 0.310 | 0.476 | 0.602 | 0.448 | 0.578 | 0.205 | 0.359 | 0.137 | 0.297 | 0.431 | 0.564 | 0.175 | 0.327 | 0.134 | 0.291 | 0.181 | 0.340 |
| | AVG | 0.101 | 0.249 | 0.189 | 0.332 | 0.179 | 0.332 | 0.116 | 0.268 | 0.380 | 0.525 | 0.538 | 0.632 | 0.168 | 0.321 | 0.123 | 0.278 | 0.343 | 0.492 | 0.125 | 0.276 | 0.121 | 0.273 | 0.168 | 0.328 |
| ETTh2 | 96 | 0.195 | 0.343 | 0.318 | 0.440 | 0.560 | 0.637 | 0.202 | 0.350 | 0.293 | 0.427 | 0.403 | 0.492 | 0.547 | 0.605 | 0.361 | 0.472 | 0.285 | 0.420 | 0.653 | 0.651 | 0.239 | 0.386 | 0.384 | 0.493 |
| | 192 | 0.240 | 0.386 | 0.362 | 0.481 | 1.228 | 0.980 | 0.254 | 0.404 | 0.316 | 0.445 | 0.536 | 0.568 | 0.920 | 0.827 | 0.501 | 0.570 | 0.312 | 0.445 | 0.313 | 0.444 | 0.275 | 0.420 | 0.498 | 0.567 |
| | 336 | 0.302 | 0.439 | 0.356 | 0.474 | 0.788 | 0.755 | 0.308 | 0.444 | 0.358 | 0.479 | 0.562 | 0.592 | 0.419 | 0.530 | 0.603 | 0.635 | 0.366 | 0.484 | 0.307 | 0.443 | 0.452 | 0.544 | 0.641 | 0.657 |
| | 720 | 0.339 | 0.467 | 0.432 | 0.528 | 1.071 | 0.891 | 0.420 | 0.521 | 0.409 | 0.513 | 0.661 | 0.652 | 0.598 | 0.629 | 1.430 | 1.043 | 0.410 | 0.513 | 0.328 | 0.458 | 0.883 | 0.797 | 1.179 | 0.926 |
| | AVG | 0.269 | 0.409 | 0.367 | 0.481 | 0.912 | 0.816 | 0.296 | 0.430 | 0.344 | 0.466 | 0.540 | 0.576 | 0.621 | 0.648 | 0.723 | 0.680 | 0.343 | 0.465 | 0.400 | 0.499 | 0.462 | 0.537 | 0.676 | 0.661 |
| ETTm1 | 96 | 0.035 | 0.143 | 0.049 | 0.170 | 0.060 | 0.187 | 0.039 | 0.151 | 0.314 | 0.492 | 0.625 | 0.655 | 0.053 | 0.176 | 0.035 | 0.144 | 0.204 | 0.381 | 0.044 | 0.159 | 0.040 | 0.152 | 0.049 | 0.170 |
| | 192 | 0.051 | 0.175 | 0.077 | 0.214 | 0.074 | 0.211 | 0.054 | 0.179 | 0.324 | 0.495 | 0.476 | 0.559 | 0.059 | 0.190 | 0.051 | 0.172 | 0.224 | 0.396 | 0.054 | 0.178 | 0.074 | 0.211 | 0.073 | 0.211 |
| | 336 | 0.067 | 0.202 | 0.102 | 0.249 | 0.100 | 0.244 | 0.071 | 0.207 | 0.317 | 0.478 | 0.458 | 0.542 | 0.106 | 0.254 | 0.067 | 0.203 | 0.237 | 0.399 | 0.095 | 0.242 | 0.106 | 0.257 | 0.099 | 0.247 |
| | 720 | 0.091 | 0.235 | 0.154 | 0.313 | 0.099 | 0.249 | 0.098 | 0.248 | 0.344 | 0.491 | 0.394 | 0.507 | 0.101 | 0.253 | 0.092 | 0.242 | 0.267 | 0.422 | 0.099 | 0.246 | 0.112 | 0.267 | 0.139 | 0.297 |
| | AVG | 0.061 | 0.189 | 0.096 | 0.237 | 0.083 | 0.223 | 0.065 | 0.196 | 0.325 | 0.489 | 0.488 | 0.566 | 0.080 | 0.231 | 0.061 | 0.190 | 0.233 | 0.400 | 0.073 | 0.206 | 0.083 | 0.222 | 0.090 | 0.231 |
| ETTm2 | 96 | 0.129 | 0.271 | 0.317 | 0.449 | 0.151 | 0.292 | 0.150 | 0.290 | 0.216 | 0.348 | 0.168 | 0.311 | 0.144 | 0.286 | 0.212 | 0.360 | 0.233 | 0.370 | 0.244 | 0.377 | 0.192 | 0.344 | 0.224 | 0.373 |
| | 192 | 0.159 | 0.304 | 0.215 | 0.358 | 0.383 | 0.527 | 0.179 | 0.325 | 0.248 | 0.387 | 0.223 | 0.366 | 0.170 | 0.319 | 0.242 | 0.376 | 0.262 | 0.399 | 0.172 | 0.314 | 0.261 | 0.399 | 0.233 | 0.379 |
| | 336 | 0.187 | 0.335 | 0.280 | 0.417 | 0.304 | 0.441 | 0.205 | 0.350 | 0.285 | 0.418 | 0.252 | 0.393 | 0.230 | 0.374 | 0.269 | 0.421 | 0.301 | 0.429 | 0.189 | 0.336 | 0.214 | 0.362 | 0.248 | 0.392 |
| | 720 | 0.230 | 0.376 | 0.321 | 0.443 | 0.544 | 0.613 | 0.289 | 0.424 | 0.323 | 0.452 | 0.314 | 0.439 | 0.254 | 0.401 | 0.270 | 0.540 | 0.336 | 0.463 | 0.259 | 0.397 | 0.258 | 0.402 | 0.292 | 0.426 |
| | AVG | 0.176 | 0.321 | 0.283 | 0.417 | 0.345 | 0.468 | 0.206 | 0.347 | 0.268 | 0.401 | 0.239 | 0.377 | 0.200 | 0.345 | 0.248 | 0.424 | 0.283 | 0.415 | 0.216 | 0.356 | 0.231 | 0.377 | 0.249 | 0.392 |
| ECL | 96 | 0.515 | 0.560 | 0.822 | 0.712 | 1.185 | 0.892 | 1.018 | 0.773 | 0.570 | 0.570 | 0.609 | 0.605 | 0.752 | 0.666 | 1.711 | 1.061 | 0.698 | 0.636 | 0.729 | 0.652 | 0.821 | 0.706 | 0.941 | 0.763 |
| | 192 | 0.518 | 0.550 | 0.741 | 0.687 | 0.687 | 0.649 | 0.923 | 0.739 | 0.580 | 0.573 | 0.707 | 0.651 | 0.801 | 0.686 | 1.141 | 0.830 | 0.725 | 0.645 | 0.953 | 0.748 | 0.779 | 0.680 | 0.994 | 0.779 |
| | 336 | 0.595 | 0.588 | 0.622 | 0.600 | 0.647 | 0.611 | 0.724 | 0.644 | 0.609 | 0.591 | 0.674 | 0.630 | 0.758 | 0.673 | 0.680 | 0.631 | 0.748 | 0.654 | 0.856 | 0.711 | 0.862 | 0.719 | 0.781 | 0.686 |
| | 720 | 0.648 | 0.621 | 0.675 | 0.636 | 0.833 | 0.712 | 1.666 | 1.041 | 0.718 | 0.644 | 0.658 | 0.627 | 0.943 | 0.781 | 0.866 | 0.727 | 0.818 | 0.695 | 0.872 | 0.719 | 0.788 | 0.690 | 0.829 | 0.707 |
| | AVG | 0.569 | 0.580 | 0.715 | 0.659 | 0.838 | 0.716 | 1.083 | 0.799 | 0.619 | 0.594 | 0.662 | 0.628 | 0.813 | 0.701 | 1.099 | 0.812 | 0.747 | 0.657 | 0.853 | 0.708 | 0.813 | 0.699 | 0.886 | 0.734 |
| Traffic | 96 | 0.489 | 0.492 | 0.663 | 0.609 | 0.490 | 0.491 | 0.479 | 0.490 | 1.514 | 1.081 | 1.334 | 1.008 | 0.490 | 0.493 | 0.784 | 0.674 | 1.436 | 1.030 | 0.803 | 0.667 | 1.377 | 0.964 | 1.532 | 0.972 |
| | 192 | 0.482 | 0.484 | 0.547 | 0.549 | 0.644 | 0.612 | 0.596 | 0.551 | 1.578 | 1.064 | 1.440 | 1.061 | 0.495 | 0.507 | 0.967 | 0.771 | 1.548 | 1.079 | 0.681 | 0.603 | 0.817 | 0.714 | 1.592 | 0.986 |
| | 336 | 0.496 | 0.503 | 0.526 | 0.541 | 0.410 | 0.443 | 0.626 | 0.566 | 1.499 | 1.056 | 1.393 | 1.040 | 0.453 | 0.491 | 0.936 | 0.748 | 1.559 | 1.093 | 0.937 | 0.733 | 1.038 | 0.812 | 1.457 | 0.944 |
| | 720 | 0.459 | 0.477 | 0.605 | 0.580 | 0.477 | 0.483 | 0.708 | 0.605 | 1.606 | 1.112 | 1.452 | 1.056 | 0.470 | 0.503 | 0.977 | 0.773 | 1.678 | 1.131 | 0.630 | 0.582 | 0.875 | 0.732 | 1.806 | 1.059 |
| | AVG | 0.482 | 0.489 | 0.585 | 0.570 | 0.505 | 0.507 | 0.602 | 0.553 | 1.549 | 1.078 | 1.405 | 1.041 | 0.477 | 0.498 | 0.916 | 0.741 | 1.555 | 1.083 | 0.763 | 0.646 | 1.027 | 0.805 | 1.597 | 0.990 |
| Weather | 96 | 0.002 | 0.028 | 0.009 | 0.075 | 0.007 | 0.065 | 0.014 | 0.087 | 0.815 | 0.778 | 2.336 | 1.354 | 0.003 | 0.041 | 0.003 | 0.043 | 0.909 | 0.829 | 0.005 | 0.058 | 0.005 | 0.052 | 0.006 | 0.062 |
| | 192 | 0.002 | 0.032 | 0.004 | 0.046 | 0.007 | 0.064 | 0.004 | 0.049 | 0.738 | 0.733 | 2.137 | 1.295 | 0.004 | 0.051 | 0.008 | 0.071 | 0.783 | 0.760 | 0.004 | 0.048 | 0.008 | 0.070 | 0.004 | 0.050 |
| | 336 | 0.002 | 0.035 | 0.004 | 0.053 | 0.013 | 0.092 | 0.011 | 0.086 | 0.731 | 0.730 | 3.159 | 1.575 | 0.004 | 0.048 | 0.011 | 0.084 | 0.850 | 0.798 | 0.006 | 0.061 | 0.005 | 0.058 | 0.006 | 0.065 |
| | 720 | 0.003 | 0.040 | 0.006 | 0.062 | 0.008 | 0.073 | 0.008 | 0.073 | 0.684 | 0.707 | 2.475 | 1.398 | 0.004 | 0.051 | 0.006 | 0.060 | 0.790 | 0.767 | 0.005 | 0.056 | 0.029 | 0.132 | 0.005 | 0.058 |
| | AVG | 0.002 | 0.034 | 0.006 | 0.059 | 0.009 | 0.073 | 0.009 | 0.074 | 0.742 | 0.737 | 2.527 | 1.406 | 0.004 | 0.048 | 0.007 | 0.065 | 0.833 | 0.789 | 0.005 | 0.056 | 0.012 | 0.078 | 0.005 | 0.059 |
| $1^{st}$ Count | | 31 | 31 | 0 | 0 | 1 | 1 | 1 | 1 | 0 | 0 | 0 | 0 | 1 | 0 | 4 | 1 | 0 | 0 | 1 | 1 | 0 | 0 | 0 | 0 |

## O  LIMITATION

The limitations of TimeSeed can be categorized into three main aspects:

**Dataset scope limitations**: TimeSeed has not been evaluated on a broader range of datasets, which hinders a comprehensive validation of its generalizability across diverse scenarios.

**Performance gap in specific scenarios**: TimeSeed exhibits a small MSE gap compared to iTransformer on the Traffic dataset. This may be attributed to its relatively small number of parameters.

**Sensitivity to outliers**: TimeSeed relies on exogenous variables to reconstruct the historical sequence of endogenous variables for forecasting future values. When exogenous variables contain a high proportion of outliers, the reconstruction becomes unstable, ultimately compromising prediction accuracy.

To address these limitations, future work could explore evaluating on more diverse and larger datasets, and adaptively scaling the model's parameter size in a controlled manner to enhance predictive performance. Additionally, incorporating outlier detection and mitigation techniques to preprocess exogenous variables could enhance their quality and the model's overall robustness. While TimeSeed achieves accurate forecasting in scenarios with missing endogenous variables, forecasting in scenarios with completely missing endogenous data remains an open research challenge.

## P  LLM USAGE

In accordance with the conference policy on large language models (LLMs), we declare that LLMs were only used as auxiliary tools to refine the grammar and fluency of sentences. No part of the research ideation, experimental design, analysis, or substantive writing was generated by LLMs.

