# OpenReview forum: "TimeSeed: Effective Time Series Forecasting with Sparse  Endogenous Variables"
_ICLR.cc/2026/Conference — Submitted to ICLR 2026_

### Official Review · Reviewer_b1Px · 2025-10-20

**Soundness:** 2
**Presentation:** 2
**Contribution:** 1
**Rating:** 2
**Confidence:** 4

**Summary:**

This paper introduce TimeSeed which is specifically designed for time series forecasting scenarios with sparsely observed endogenous variables. Technically, TimeSeed leverages dense exogenous and sparse endogenous sequences within a two-stage paradigm of context reconstruction and hierarchical prediction. Experiments demonstrate that TimeSeed consistently outperforms state-of-the-art models in forecasting accuracy.

**Strengths:**

1. The paper is well-organized and easy to follow.
2. The proposed TimeSeed is a lightweight model built entirely upon linear layers, which significantly reduces computational costs.
3. The experiment is extensive and quite detailed.

**Weaknesses:**

1. The paper focuses on the sparse forecasting scenario, which is a conceptually complex and challenging scenario. However, practical evidence on this setting is limited, as the experiments primarily rely on existing multivariate benchmarks.
2. The presentation of TimeSeed lacks clarity. For instance, line 162 of the main text states, "we design the TDA and FDA blocks to predict trend and periodic features for sparse endogenous sequences, respectively." In contrast, line 171 states, "the TDA is designed to learn the periodic features of endogenous variables by leveraging exogenous variables."
3. Furthermore, the paper does not provide adequate explanations or experimental validation regarding the significance and effectiveness of decomposing and modeling the time domain and frequency domain separately.
4. The term "physical similarity" in line 53 is unclear in the context of time series forecasting. A more precise explanation is required.

**Questions:**

1. Why did the authors choose to introduce a two-stage decomposition and forecasting paradigm based on reconstruction instead of directly predicting the future of endogenous variables? The paper lacks detailed explanations and experimental evidence to justify this choice.
2. In Figure 3 (right), the authors used the Pearson coefficient to estimate the correlation between the reconstructed sequences and the ground truth. Why did the authors choose the Pearson coefficient instead of Dynamic Time Warping (DTW), which is generally more suitable for time series data?

---

> ### Author Response · Authors · 2025-11-25
> **Response to Reviewer b1Px**
>
> > W1: Find real-world benchmark.
>
> We have incorporated the real clinical dataset $\underline{PhysioNet}$, where physiological variables naturally exhibit sparsity. We use reliably obtainable signals such as HR, RespRate, Temp, SysABP, DiasABP, and MAP as exogenous variables, while the more sparsely observed Glucose serves as the endogenous variable. We use the first 24 hours of observations to forecast the subsequent 24 hours. As shown in Table, TimeSeed still achieves the best or highly competitive performance under these genuinely sparse conditions, demonstrating the robustness and practical applicability of our method.
>
> |Model|TimeSeed||DUET||DLinear||TimeXer||FilterNet||SparseTSF||
> |:-:|:-:|:--:|:-:|:-:|:-:|:-:|:-:|:-:|:-:|:-:|:-:|:-:|
> |Metric|MSE|MAE|MSE|MAE|MSE|MAE|MSE|MAE|MSE|MAE|MSE|MAE|
> |PhysioNet|0.27|0.20|0.34|0.24|0.33|0.24|0.78|0.53|0.33|0.24|0.35|0.25|
>
>
>
> > W2: The presentation of TimeSeed lacks clarity.
>
> We will adopt this suggestion in the revised version of the paper. Specifically, the modifications will include changing line 162 and similar statements from “predict trend and periodic features” to “**reconstruct** the trend and periodic components.”
>
>
>
> > W3: Demonstrate the effectiveness of decomposition in the time and frequency domains.
>
> **Experience:** Decomposing complex problems into more manageable subproblems is a widely recognized strategy. As stated in $\underline{Section 3.2}$, transforming sparse prediction tasks into sequence-based prediction tasks can effectively enhance robustness. Empirically, this form of decomposition is reasonable.
>
> **Experiments:** In $\underline{Section 4.3}$, results show that removing either FDA or TDA leads to a significant performance drop. For example, on the ETTh2 dataset, removing FDA increases MSE from 0.272 to 0.299 (**a 9.9% performance decline**), while removing TDA raises MSE to 0.376 (**a 38.2% decline**). The ablation studies clearly demonstrate the contribution of the frequency-domain and time-domain components to overall prediction accuracy.
>
> In $\underline{Section 4.6}$, Figure 5(a) shows that on the ETTh1 dataset, the reconstructed trend component almost completely overlaps with the true trend in the frequency domain, indicating that TimeSeed can accurately extract endogenous trend information from exogenous inputs.
>
> Also in$\underline{Section 4.6}$, Figure 5(c) reports the Pearson correlation coefficients between the reconstructed sequences and the ground truth. **The average correlation coefficients for the endogenous components reconstructed by TDA, FDA, and ASR with the true values are 0.532, 0.885, and 0.5, respectively.**
>
>
>
> > W4: Explain the term “physical similarity” in line 53 of the main text.
>
> Motrenko et al., in *“Combining endogenous and exogenous variables in a special case of non-parametric time series forecasting model”*, systematically elaborated on the physical relationships between endogenous and exogenous variables, highlighting that in domains such as industrial systems and environmental monitoring, variables are coupled through physics-based mechanisms.
>
> Huang et al., in *“Exploiting language power for time series forecasting with exogenous variables”*, distinguished between “semantic similarity” and “physical similarity” among variables, emphasizing that physical similarity specifically refers to coordinated variation patterns arising from variables being governed by the same physical laws over time.
>
> Pandit et al., in *“Hybrid time series models with exogenous variable for improved yield forecasting of major Rabi crops in India”*, demonstrated through agricultural case studies that there exists a quantifiable physical dependency between exogenous variables (e.g., temperature, precipitation) and endogenous variables (crop yield).
>
> Lu et al., in *“CATS: Enhancing multivariate time series forecasting by constructing auxiliary time series as exogenous variables”*, showed that different variables within the same physical system share similar energy distribution characteristics in the frequency domain.
>
> **A considerable body of literature supports this concept, and we have added the relevant citations in the revised version.**

---

> ### Author Response · Authors · 2025-11-25
> **Response to Reviewer b1Px**
>
> > Q1: Why not directly predicting the future of endogenous variables?
>
> We have incorporated your suggestion into the ablation experiments for TimeSeed. In the table below, the two-stage decomposition and forecasting paradigm outperforms the approach of directly predicting endogenous variables. In particular, on ETTh2 and ETTm2, TimeSeed shows significant improvements in both MSE and MAE, indicating that the model is more robust in capturing trend and periodic features. This is because the two-stage approach decouples sequence features, allowing the reconstruction stage to focus more on trends and periodic components, thereby enhancing robustness.
>
> The complete results are shown in **Table 11** of $\underline{Appendix H}$.
>
> |Datasets||ETTh1||ETTm1||ETTh2||ETTm2||
> |:-:|:-:|:-:|:-:|:-:|:-:|:-:|:-:|:-:|:-:|
> |Metric||MSE|MAE|MSE|MAE|MSE|MAE|MSE|MAE|
> |Ours||**0.096**|**0.242**|**0.068**|**0.194**|**0.272**|**0.410**|**0.149**|**0.289**|
> |Direct Forecasting||0.124|0.274|0.074|0.204|0.400|0.493|0.160|0.300|
>
>
>
> > Q2: Why not choose Dynamic Time Warping  in Figure 3 (right).
>
> **We didn’t have any particularly deep reason for choosing the Pearson correlation coefficient**. We just wanted a simple and intuitive metric to show the correlation between the reconstructed and ground-truth sequences. Following your suggestion, we reran the case study code, updated Figure 3 (right), and reported the DTW metric.

---

> ### Comment · Reviewer_b1Px · 2025-11-27
>
> Thank you for the detailed response. All my concerns have been resolved, and I will increase my score.

---

### Official Review · Reviewer_eg86 · 2025-10-31

**Soundness:** 3
**Presentation:** 3
**Contribution:** 2
**Rating:** 6
**Confidence:** 3

**Summary:**

This paper introduces TimeSeed, a model for time series forecasting under the specific and challenging setting where the target (endogenous) variable is sparsely observed, while correlated (exogenous) variables are fully observed. The proposed method operates in a "reconstruct-then-predict" fashion. It first generates a complete historical representation of the target series by aggregating information from three modules: a Time Domain Aggregator (TDA) for periodic patterns, a Frequency Domain Aggregator (FDA) for trend information (both derived from exogenous series), and an Adaptive Scale Reconstructor (ASR) which incorporates the sparse endogenous signal itself. This reconstructed series is then fed into a simple linear forecaster. The authors show that this efficient, linear-based model achieves state-of-the-art results on seven benchmark datasets under this sparse setting, outperforming numerous recent and complex models.

**Strengths:**

1) The paper formalizes and tackles the f(X, S) -> Y problem, a relevant and common scenario in real-world applications where a target signal is costly or difficult to measure continuously.

2) Strong Empirical Results: Within its defined experimental setup, TimeSeed demonstrates a significant and consistent performance advantage over a wide range of strong baselines.

3) The model is extremely lightweight and computationally efficient, making it a practical and attractive solution for deployment.

4) The authors' commitment to re-running all baselines under a fair and unified framework is a major strength.

**Weaknesses:**

1) The main weakness is the limited conceptual novelty. The model is largely a thoughtful recombination of existing ideas (patching, FFT) from recent time series literature, applied to a new problem variant. The contribution feels more like system-building than the introduction of a new fundamental modeling principle.

2) The central "reconstruction" narrative is not backed by an explicit reconstruction loss, which makes the model's inner workings less interpretable and the claims less grounded than they appear. The model is not learning to reconstruct the past per se, but rather learning a useful latent representation for forecasting.

3) The evaluation is almost entirely based on a uniform sparsity pattern, which is not representative of many real-world missing data scenarios (e.g., random or block-wise missingness). The model's robustness to more realistic data imperfection is not sufficiently demonstrated.

4) The paper does not compare against a straightforward two-stage pipeline of "imputation model + forecasting model." This baseline is essential for determining if the proposed integrated architecture provides a tangible benefit over a simple, modular approach.

**Questions:**

Could you clarify the design choice of not using a direct reconstruction loss on the historical window? Given the paper's central narrative is "reconstruction," it seems this would not only align the training objective with the story but also provide a more grounded way to supervise and evaluate the TDA, FDA, and ASR modules.

The ASR module uses Gumbel-Softmax for a hard selection of a single resolution. What is the justification for this restrictive choice over a soft, weighted combination of features from all reconstructed scales (O_q), which would be a more general approach and potentially capture information across multiple resolutions simultaneously?

A critical baseline seems to be missing: a two-stage approach where a state-of-the-art imputation model (e.g., SAITS, or even a masked version of a model like PatchTST) is first used to fill the sparse endogenous series, followed by a separate state-of-the-art forecasting model. How do you expect TimeSeed to perform against such a pipeline? This comparison is crucial to validate the benefits of your integrated "reconstruct-then-predict" architecture.

How does the model's performance degrade as the sparsity pattern becomes more challenging, for example, with large, contiguous blocks of missing data instead of uniform samples? The current reliance on TDA/FDA might be robust, but this needs to be empirically verified to claim general applicability.

---

> ### Author Response · Authors · 2025-11-25
> **Response to Reviewer eg86**
>
> > W1: TimeSeed integrates multiple techniques to build a unified system. Provide a detailed description of its conceptual innovations.
>
> Thank you. Unlike existing methods that mainly focus on **“directly mapping inputs to future values,”** the core idea of TimeSeed lies in **fully leveraging complete exogenous variables alongside extremely sparse endogenous observations**. Since current forecasting models (e.g., DLinear) are already quite mature, our goal is no longer to redesign the predictor, but to **reconstruct a complete and reliable historical endogenous sequence as accurately as possible**, thereby ensuring competitiveness in the subsequent forecasting stage.
>
> Following this approach, we transform the originally highly challenging **“sparse forecasting problem”** into a more robust **“sequence forecasting problem.”** This **two-stage “reconstruction–forecasting ” paradigm represents a paradigm shift rather than a simple stacking of modules.** It emphasizes the use of stable information from exogenous variables to compensate for missing endogenous data, an approach that has not been systematically explored in previous research.
>
> Therefore, we argue that the novelty of TimeSeed does not lie in the mere combination of components, but in the **entirely new paradigm, the motivation behind the design, and its logical structure.** For example, decomposing trend and seasonality enhances the robustness of the reconstruction stage, while FFT helps the model capture fine-grained structures beyond the main trend. Compared with simply pursuing **“flashier or more novel” techniques** to achieve performance gains, our focus is on establishing **a fundamentally new forecasting paradigm tailored to scenarios with sparse endogenous variables.** This is precisely the intention and contribution of TimeSeed.
>
>
>
> > W2 & Q1：Why not use direct reconstruction loss. The model is not learning to reconstruct the past itself; rather, it is learning a useful latent representation for prediction.
>
> When initially designing TimeSeed, we did not include an explicit reconstruction loss. Even without hyperparameter tuning, **we found that the model could already achieve competitive performance.** While introducing a reconstruction loss can bring some additional gains, the associated extra gradient computations would undermine the model’s lightweight nature and make comparisons with other baselines less fair.
>
> We agree with your point: the model’s objective is not to reconstruct past sequences per se, but to learn latent representations that are useful for prediction tasks. **Prior studies $\underline{(e.g., TimeBase, ICML 2025)}$ have shown that relying solely on the effective latent representations contained in sequences is sufficient to achieve strong predictive performance.** Given the physical correlations between exogenous and endogenous variables, the model only needs to learn a useful latent endogenous representation from the rich exogenous information and the sparse endogenous observations.
>
> Although we do not use an explicit reconstruction loss, we have rigorously validated the effectiveness and interpretability of the reconstruction process. $\underline{Section 4.6}$ provides related reliability analyses. In **Figure 5 (right)**, we report the correlation between the reconstructed representations and the true historical sequences: the Pearson correlation coefficients between the reconstructed outputs of TDA and FDA and the ground truth are 0.532 and 0.885, respectively. Additionally, **Figures 5(a) and (b)** visualize the frequency spectra and weight distributions, further illustrating how the model captures temporal structures. While introducing an explicit reconstruction loss could improve model transparency, it would also increase complexity and computational cost, deviating from our goal of building a lightweight model.
>
> TimeSeed achieves significant predictive performance improvements across multiple datasets through end-to-end training (**average MSE reduced by 13.01%**), demonstrating that its implicit reconstruction mechanism is effective even when integrated within the prediction task.

---

> ### Author Response · Authors · 2025-11-25
> **Response to Reviewer eg86**
>
> > W3: Robustness should be evaluated under real-world data missing scenarios, such as random missingness or block missingness.
>
> Good. We have incorporated the real clinical dataset $\underline{PhysioNet}$, where physiological variables naturally exhibit sparsity. We use reliably obtainable signals such as HR, RespRate, Temp, SysABP, DiasABP, and MAP as exogenous variables, while the more sparsely observed Glucose serves as the endogenous variable. We use the first 24 hours of observations to forecast the subsequent 24 hours. As shown in Table, TimeSeed still achieves the best or highly competitive performance under these genuinely sparse conditions, demonstrating the robustness and practical applicability of our method.
>
> |Model|TimeSeed||DUET||DLinear||TimeXer||FilterNet||SparseTSF||
> |:-:|:-:|:--:|:-:|:-:|:-:|:-:|:-:|:-:|:-:|:-:|:-:|:-:|
> |Metric|MSE|MAE|MSE|MAE|MSE|MAE|MSE|MAE|MSE|MAE|MSE|MAE|
> |PhysioNet|0.27|0.20|0.34|0.24|0.33|0.24|0.78|0.53|0.33|0.24|0.35|0.25|
>
> In $\underline{Appendix G}$, we specifically report the complete experimental results under the **random sampling strategy**. The results show that, although random missingness introduces higher uncertainty, causing a slight performance drop for all models, TimeSeed still maintains a significant competitive advantage. For example, on the ETTh2 dataset, TimeSeed achieves an average MSE of 0.278 under random sampling, still outperforming baseline models such as DLinear (0.402) and TimeXer (0.339), demonstrating TimeSeed’s generalization capability to irregular missing patterns.
>
> |Model|TimeSeed||TimeSeed*||DLinear||DLinear*||TimeXer||TimeXer*||FITS||FITS*||
> |:-:|:-:|:-:|:-:|:-:|:-:|:-:|:-:|:-:|:-:|:-:|:-:|:-:|:-:|:-:|:-:|:-:|
> |Metric|MSE|MAE|MSE|MAE|MSE|MAE|MSE|MAE|MSE|MAE|MSE|MAE|MSE|MAE|MSE|MAE|
> |ETTh1|0.096|0.242|0.121|0.267|0.113|0.262|0.114|0.263|0.444|0.569|0.527|0.629|0.142|0.297|0.142|0.296|
> |ETTh2|0.272|0.410|0.278|0.415|0.409|0.493|0.402|0.489|0.342|0.465|0.339|0.463|0.553|0.587|0.566|0.595|
> |ETTm1|0.068|0.194|0.070|0.196|0.075|0.205|0.078|0.210|0.344|0.499|0.351|0.504|0.079|0.216|0.079|0.216|
> |ETTm2|0.149|0.289|0.170|0.314|0.165|0.303|0.196|0.337|0.266|0.404|0.273|0.406|0.202|0.343|0.269|0.402|
>
> Moreover, in $\underline{Appendix I}$, we explore a more challenging extreme scenario: **forecasting using only exogenous variables** (with endogenous variables completely missing). The experimental results show that some models, such as TimeMixer, suffer severe performance degradation under this setting, whereas TimeSeed, leveraging its two-stage paradigm for reconstructing historical sequences, demonstrates remarkable robustness and is able to produce reasonable predictions. This further illustrates that our model’s unique design enhances its tolerance to incomplete data.
>
> |Model|TimeSeed||iTrans||TimeXer||TimeMixer||
> |:-:|:-:|:-:|:-:|:-:|:-:|:-:|:-:|:-:|
> |Metric|MSE|MAE|MSE|MAE|MSE|MAE|MSE|MAE|
> |Traffic|0.424|0.453|0.619|0.587|0.633|0.680|0.591|0.652|
> |Weather|0.007|0.069|0.009|0.076|0.801|0.762|2.227|1.317|
>
> To further evaluate the model’s sensitivity to changes in data scale, in the main text $Section 4.5$ Different Data Scale, we systematically reduced the available information, **including sparser endogenous sequences, fewer exogenous variables, and shorter historical windows**. Under these more challenging configurations, TimeSeed’s performance degradation is far smaller than that of DUET. For example, when the number of exogenous variables is reduced, TimeSeed’s MSE increases by only 18.84%, whereas DUET’s MSE deteriorates by 126.61%, strongly demonstrating the strong generalization and robustness of our method under information-limited conditions.
>
> Additionally, in $Appendix J$, we provide error analyses and statistical test results from repeated experiments, showing that TimeSeed’s performance gains are stable and significant. Its **minimal standard deviations** further reinforce the reliability of our conclusions.
>
>
> |Model|TimeSeed||Confidence|
> |:-:|:-:|:-:|:-:|
> |Dataset|MSE|MSE|Interval|
> |Weather|0.002 ± 0.002|0.035 ± 0.007|0.99|
> |ECL|0.533 ± 0.035|0.560 ± 0.019|0.99|
> |Traffic|0.404 ± 0.017|0.435 ± 0.014|0.99|
> |ETTh1|0.097 ± 0.003|0.243 ± 0.005|0.99|
> |ETTh2|0.271 ± 0.004|0.409 ± 0.003|0.99|
> |ETTm1|0.068 ± 0.003|0.194 ± 0.004|0.99|
> |ETTm2|0.148 ± 0.002|0.289 ± 0.001|0.99|

---

> ### Author Response · Authors · 2025-11-25
> **Response to Reviewer eg86**
>
> > Q2: Why not choose weighted combination of features from all reconstructed scales？
>
> In scenarios with extremely sparse data, the information quality and reliability of upsampled sequences at different resolutions vary significantly. A simple soft-weighted combination (e.g., a weighted average over all *$O_q$*) may cause low-quality or noisy scale features to **dilute or interfere** with the contribution of high-quality scale features. For example, the Case Study in **Figure 3 (right)** reports the resolution selection on ETTh1. The lowest selection weight is 0.3, and the highest is approximately 0.6. By using Gumbel-Softmax to constrain the model to **lock onto the most valuable resolution scale each time**, the influence of noise can be reduced.
>
> The hard-selection mechanism also helps **control model complexity and prevent overfitting**. For a lightweight model with only 0.19M parameters, forcing the model to focus on a single scale is a more efficient and robust strategy. This is particularly beneficial under limited data conditions, as it encourages the model to capture more fundamental patterns rather than noise.
>
> To address your comment, we conducted a comparative experiment between soft-weight and hard-weight selection. The results show that the hard-selection strategy outperforms soft-weight combinations across all datasets. This is likely because hard selection avoids interference from low-quality scale information in the final representation, thereby reducing the risk of overfitting.
>
> |Model|TimeSeed||TimeSeed(Soft)||
> |-|:-:|:-:|:-:|:-:|
> |Metric|MSE|MAE|MSE|MAE|
> |ETTh1|0.096|0.242|0.119|0.266|
> |ETTm1|0.068|0.194|0.092|0.226|
> |Traffic|0.403|0.433|0.545|0.567|
> |Weather|0.002|0.034|0.009|0.071|

---

> ### Author Response · Authors · 2025-11-28
> **Response to Reviewer eg86**
>
> > W4 &Q3: Need to use a state-of-the-art imputation model to fill in the sparse endogenous series, and then apply another state-of-the-art forecasting model.
>
> Good. We first employ PatchTST to impute the sparse endogenous series. After obtaining the complete sequences, another state-of-the-art forecasting model (PatchTST or TimeXer) is applied to predict future values. As shown in the table, TimeSeed achieves superior performance on most datasets. On average, compared with PatchTST/PatchTST, TimeSeed reduces MSE and MAE by approximately 36% and 24%, respectively, and compared with PatchTST/TimeXer, it reduces MSE and MAE by approximately 26% and 30%, respectively. This improvement arises from the two-stage approach, which decouples the sequence features and allows the reconstruction stage to focus more on capturing trends and periodic patterns, thereby enhancing robustness.
>
> The limited contribution of the imputation model may be due to the extreme sparsity of the endogenous series, and introducing an additional imputation model increases the number of parameters, which in turn raises the risk of overfitting.
>
> |Model|TimeSeed||PatchTST/PatchTST||PatchTST/TimeXer||
> |:-:|:-:|:-:|:-:|:-:|:-:|:-:|
> |Metric|MSE|MAE|MSE|MAE|MSE|MAE|
> |ETTh1|0.096|0.242|0.153|0.322|0.388|0.498|
> |ETTh2|0.272|0.410|0.502|0.561|0.281|0.412|
> |ETTm1|0.068|0.194|0.041|0.196|0.283|0.428|
> |ETTm2|0.149|0.289|0.154|0.293|0.197|0.336|

---

### Official Review · Reviewer_e8px · 2025-11-01

**Soundness:** 2
**Presentation:** 3
**Contribution:** 2
**Rating:** 4
**Confidence:** 3

**Summary:**

The paper presents TimeSeed, a simple yet effective framework for forecasting when the target variable is only sparsely observed. It rebuilds a dense context of the target using exogenous variables and the few available target points, then forecasts normally. The model combines three parts: 1) Time-Domain Aggregator captures periodic patterns; 2) Frequency-Domain Aggregator extracts trend information using FFTs; 3) Adaptive Scale Reconstructor upsamples sparse data at multiple resolutions. TimeSeed is fully linear, very lightweight, and achieves improvements over baselines across seven datasets.

**Strengths:**

1. TimeSeed presents a effective approach to forecasting with sparse target data. It reframes the problem as a reconstruction-forecasting task, using simple linear modules that separate periodic and trend information through time- and frequency-domain modeling.

2. The paper is clearly written and supported by various experiments, showing consistent gains on ablations that validate each component.

3. The model’s clarity, lightweight design, and efficiency make it practical for real-world deployment, while its empirical results show contribution for scenarios with limited target observations.

**Weaknesses:**

1. The authors did not use real datasets with sparsely observed endogenous variables and fully available exogenous variables. Instead, they created such conditions artificially by applying predefined sparsity ratios to general datasets that originally contain complete endogenous and exogenous information. This design choice somewhat weakens the paper’s practical significance. To better demonstrate the real-world relevance of this problem, it would be valuable for the authors to identify or include datasets that naturally reflect this sparse-endogenous scenario.

2. How exactly is the sparsity ratio implemented? Providing a detailed description of how the sparse endogenous variables are constructed from the full datasets would help alleviate concerns about selection bias and fairness in comparison. When the sparsity ratio is high, different random seeds could substantially alter the time series behavior (e.g. especially for data with sparse-peak patterns). The impact of such randomness and its implications for statistical significance should be discussed more thoroughly.

3. Additionally, were the baselines trained on the imputed, non-observed timesteps? Training models on data points known to be inaccurate could cause them to learn false inter-series relationships, which would unfairly hurt the performance of the models that focus on learning precise inter-series dependencies.

**Questions:**

1. In Tables 1 and 2, it seems unusual that DLinear outperforms almost all other baselines, given that it does not take exogenous variables as input and has very few parameters. This observation reinforces the concern raised in Weakness 3 about how the baselines are adapted or trained. Moreover, the setup of the experiment in Table 2 needs clearer explanation: in Section 4.1, the paper states that “endogenous variables are missing.” In that case, what exactly does a DLinear-type model use as input?

2. Is the Adaptive Scale Reconstructor reconstructing the fully dense time series, or only the sparsely observed parts? How is the reconstruction quality of the historical series evaluated. Does the model apply a reconstruction loss on the full series?

3. For input lengths longer than 96, how does extending the context window affect performance? It would be helpful if the authors could include additional experiments or discussion on this aspect.

If the authors' response adequately addresses my questions and concerns mentioned above, I am willing to raise my score.

---

> ### Author Response · Authors · 2025-11-25
> **Response to Reviewer e8px**
>
> > W1: Find a dataset with real-world scenarios
>
> Good. We have incorporated the real clinical dataset $\underline{PhysioNet}$, where physiological variables naturally exhibit sparsity. We use reliably obtainable signals such as HR, RespRate, Temp, SysABP, DiasABP, and MAP as exogenous variables, while the more sparsely observed Glucose serves as the endogenous variable. We use the first 24 hours of observations to forecast the subsequent 24 hours. As shown in Table, TimeSeed still achieves the best or highly competitive performance under these genuinely sparse conditions, demonstrating the robustness and practical applicability of our method.
>
> |Model|TimeSeed||DUET||DLinear||TimeXer||FilterNet||SparseTSF||
> |:-:|:-:|:--:|:-:|:-:|:-:|:-:|:-:|:-:|:-:|:-:|:-:|:-:|
> |Metric|MSE|MAE|MSE|MAE|MSE|MAE|MSE|MAE|MSE|MAE|MSE|MAE|
> |PhysioNet|0.27|0.20|0.34|0.24|0.33|0.24|0.78|0.53|0.33|0.24|0.35|0.25|
>
>
>
>
>
> >  W2: How the sparsity rate is implemented? What is the impact of the random seed on its application?
>
> **Regarding the implementation of sparsity:** In the main text, we construct sparse endogenous sequences using **uniform interval sampling**, without involving any random seeds. Specifically, the sparsity rate is defined as the ratio of observed endogenous values to exogenous variables. This deterministic sampling strategy ensures reproducibility and eliminates selection bias when comparing different models.
>
> **Regarding concerns about randomness:** To comprehensively evaluate the model’s robustness under real-world data missing scenarios, we provide extensive experiments with **random sampling** in $\underline{Appendix \ G}$(**Tables 7 and 8** ). In these experiments, all models share the same random seeds and hyperparameters (**Table 6**) to ensure a fair comparison. The results show that, although random sampling generally degrades the performance of all baselines (consistent with the reviewer’s observation regarding sparse peak patterns), **TimeSeed consistently achieves the best performance under both sampling strategies**. This confirms the robustness of our model to different data missing patterns.
>
> |Model|TimeSeed||TimeSeed*||DLinear||DLinear*||TimeXer||TimeXer*||FITS||FITS*||
> |:-:|:-:|:-:|:-:|:-:|:-:|:-:|:-:|:-:|:-:|:-:|:-:|:-:|:-:|:-:|:-:|:-:|
> |Metric|MSE|MAE|MSE|MAE|MSE|MAE|MSE|MAE|MSE|MAE|MSE|MAE|MSE|MAE|MSE|MAE|
> |ETTh1|0.096|0.242|0.121|0.267|0.113|0.262|0.114|0.263|0.444|0.569|0.527|0.629|0.142|0.297|0.142|0.296|
> |ETTh2|0.272|0.410|0.278|0.415|0.409|0.493|0.402|0.489|0.342|0.465|0.339|0.463|0.553|0.587|0.566|0.595|
> |ETTm1|0.068|0.194|0.070|0.196|0.075|0.205|0.078|0.210|0.344|0.499|0.351|0.504|0.079|0.216|0.079|0.216|
> |ETTm2|0.149|0.289|0.170|0.314|0.165|0.303|0.196|0.337|0.266|0.404|0.273|0.406|0.202|0.343|0.269|0.402|
>
> **Statistical Significance:** In **Table 11**  $\underline{Appendix J}$, we report the mean ± standard deviation over five independent runs along with the 99% confidence intervals. The small standard deviations demonstrate the statistical reliability and reproducibility of our experimental results.
>
> |Model|TimeSeed||Confidence|
> |:-:|:-:|:-:|:-:|
> |Dataset|MSE|MSE|Interval|
> |Weather|0.002 ± 0.002|0.035 ± 0.007|0.99|
> |ECL|0.533 ± 0.035|0.560 ± 0.019|0.99|
> |Traffic|0.404 ± 0.017|0.435 ± 0.014|0.99|
> |ETTh1|0.097 ± 0.003|0.243 ± 0.005|0.99|
> |ETTh2|0.271 ± 0.004|0.409 ± 0.003|0.99|
> |ETTm1|0.068 ± 0.003|0.194 ± 0.004|0.99|
> |ETTm2|0.148 ± 0.002|0.289 ± 0.001|0.99|
>
> $\underline{Appendix I}$,we explore a more challenging extreme scenario: **Forecasting using only exogenous variables** (with endogenous variables completely missing). The results show that some models suffer severe performance degradation under this setting, whereas TimeSeed, leveraging its two-stage paradigm for reconstructing historical sequences, demonstrates remarkable robustness. This further illustrates that our model’s unique design enhances tolerance to incomplete data.
>
> |Model|TimeSeed||iTrans||TimeXer||TimeMixer||
> |:-:|:-:|:-:|:-:|:-:|:-:|:-:|:-:|:-:|
> |Metric|MSE|MAE|MSE|MAE|MSE|MAE|MSE|MAE|
> |Traffic|0.424|0.453|0.619|0.587|0.633|0.680|0.591|0.652|
> |Weather|0.007|0.069|0.009|0.076|0.801|0.762|2.227|1.317|
>
> In the **Different Data Scale** experiments in $\underline{Section 4.5}$ of the main text (Figure 4, left), we systematically reduced the available information, including sparser endogenous sequences, fewer exogenous variables, and shorter historical windows. Under these more challenging configurations, TimeSeed’s performance degrades far less than that of DUET; for instance, when exogenous variables are reduced, TimeSeed’s MSE increases by only 18.84%, whereas DUET’s MSE worsens by 126.61%. This strongly demonstrates the superior generalization and robustness of our method under information-constrained conditions.

---

> ### Author Response · Authors · 2025-11-25
> **Response to Reviewer e8px**
>
> > W3 : Was training conducted using imputed, unobserved time steps? Is this fair?
>
> **Regarding the Training Setup of Baseline Models:** In our experimental setup, the exogenous sequences are complete, while the endogenous sequences are sparse. Since the architectures of existing time series forecasting models cannot directly handle inputs of inconsistent lengths, we equipped all baseline models with **learnable upsampling layers identical to those in TimeSeed**. These layers upsample the sparse endogenous sequences to match the length of the exogenous sequences before concatenation. This ensures that all models receive inputs of the same dimensionality, maintaining a fair comparison.
>
> **Regarding Concerns About Spurious Relationship Learning:** It should be emphasized that the upsampled time steps **are not inaccurate interpolations or PAD tokens, but feature representations generated via learnable parameters**. All models are trained under the same hyperparameter settings to learn how to extract meaningful information from sparse observations. The key distinction is that TimeSeed models trends and seasonality separately from exogenous data via a decomposition-reconstruction-prediction approach and further refines predictions using the sparse endogenous series. The results demonstrate that our multi-stage strategy indeed provides more robust performance on this challenging forecasting problem.
>
> **Fairness Guarantee:** We took the following measures to ensure fair comparison: (1) all models use the same upsampling strategy and hyperparameters (Table 6); (2) all models are trained and evaluated on identical input representations; (3) we further conducted single-point experiments (Appendix M) and even experiments using only exogenous variables (Appendix I), where TimeSeed still achieved competitive results, validating its effectiveness under extreme sparsity conditions.
>
>
>
>
>
> > Q1:Unexpected performance of the linear model? What are the inputs? Describe the experimental setup in more detail.
>
> **Outstanding Performance of DLinear:** DLinear’s strong performance in sparse scenarios highlights an important finding: **under conditions of data sparsity, simple linear models often outperform more complex models**. This phenomenon aligns with conclusions from recent studies (Zeng et al., 2023; Xu et al., 2023), which show that over-parameterized models are prone to overfitting when data is limited. Our model, **TimeSeed**, continues this design philosophy, employing a lightweight linear architecture (only 0.19M parameters) and further enhancing performance through an explicit **decompose–reconstruct–predict** mechanism.
>
> **Regarding Baseline Model Input Handling:** As noted in our response to Weakness 3, all baseline models (including DLinear) use an identical input construction process: sparse endogenous sequences are first restored to the same length as the exogenous sequences via a **learnable upsampling layer**, then concatenated and fed into the model. Due to space limitations, this was briefly described in Section 4 *Implementation Details* of the main text. In the revised version, we will clarify the details of these experimental settings. We also provide baseline model code suitable for sparse prediction scenarios to ensure reproducibility.
>
> **Regarding the Misstatement in Section 4.1:** The statement in Section 4.1 that “endogenous variables are missing” is not entirely accurate. In fact, Table 2 reports results under a more extreme prediction scenario, containing only a single endogenous point. We will revise this statement in the updated version for greater accuracy.

---

> ### Author Response · Authors · 2025-11-25
> **Response to Reviewer e8px**
>
> > Q2: What does the Adaptive Scale Reconstructor reconstruct? Is a reconstruction loss applied?
>
> **On the ASR Reconstruction Objective:** The Adaptive Scale Reconstructor (ASR) is designed to reconstruct the **complete dense historical endogenous sequence** from sparse endogenous observations. Specifically, it generates sequence representations at multiple resolutions through multi-scale upsampling (Equations 9–12) and employs Gumbel Softmax to adaptively select the optimal scale for reconstruction. This process enables the model to capture dynamic patterns of the endogenous sequence at different temporal granularities.
>
> **On Evaluating Reconstruction Quality:** **(1)** Figure 3 (right) presents a representative case on the ETTh1 dataset, reporting the Pearson correlation coefficients between reconstructed sequences at different resolutions and the ground truth. The best reconstructed resolution achieves a correlation of 0.59, validating ASR’s effective selection mechanism; **(2)** Table 4 shows that removing the ASR module leads to a significant performance drop (MSE on ETTm2 increases from 0.149 to 0.176), indirectly demonstrating the reconstruction quality.
>
> **On Applying Reconstruction Loss to the Entire Sequence:** We choose not to use an explicit reconstruction loss for the following reasons: **(1)** TimeSeed demonstrates competitive performance across all datasets, indicating that implicitly learned reconstruction already satisfies prediction needs. Ablation studies further confirm the contribution of each component; **(2)** baseline models do not use auxiliary reconstruction tasks, so adding extra loss for TimeSeed would make comparisons unfair; **(3)** the ultimate goal is accurate prediction of future endogenous variables, and explicit reconstruction loss may force the model to overfit sparse observations, harming generalization. Our hierarchical prediction paradigm (Equations 13–14) implicitly optimizes reconstruction quality by combining outputs from FDA, TDA, and ASR. This end-to-end learning is more suitable for sparse scenarios.
>
>
>
>
>
> > Q3: How does a longer context window impact performance? Provide experimental results and discussion.
>
> **On the Effect of Different Input Lengths:** We have provided a detailed experimental analysis in $\underline{Appendix \ C}$ (**Figures 6 and 7**), systematically evaluating the impact of different look-back window lengths $T \in \{96, 192, 336, 528, 720\}$ on model performance. The results show that: **(1)** on the ETTm2 dataset, increasing the input length leads to a significant reduction in MSE when the prediction horizons are 192 and 336 steps; **(2)** the performance improvement remains stable across different prediction horizons, validating that our decomposition-reconstruction prediction mechanism can effectively leverage extended historical information without introducing redundancy.
>
> We will explicitly guide readers to refer to the full results in Appendix C in Section 4.4 (Model Analysis) of the revised version.

---

### Author Response · Authors · 2025-12-01
**Author Final Remarks**

Dear AC, and the reviewers,

We sincerely thank you for your invaluable expertise and dedication throughout the review process. We hope the following final remarks could clarify our paper’s contributions and summarize the rebuttals, therefore serving as a constructive reference for decision-making.

**Contributions**

* **New Prediction Paradigm:** We propose a new prediction paradigm $f(X, S) \to Y$ for scenarios with sparse endogenous variables and dense exogenous variable, a common yet often overlooked setting in real-world applications. To address this challenging prediction problem, we develop a two-stage reconstruction-then-prediction pipeline that leverages rich exogenous information to compensate for the sparsity in endogenous observations. This decomposition transforms the original sparse prediction task into a standard sequence-based forecasting problem, thereby substantially enhancing robustness.


* **Innovative Framework:** We propose TimeSeed, a lightweight model that leverages dense exogenous and sparse endogenous sequences within a two-stage paradigm of context reconstruction and hierarchical prediction. Endogenous periodic and trend components are captured via Time Domain Aggregator and Frequency Domain Aggregator , and refined with an Adaptive Scale Reconstructor.


* **State-of-the-Art Performance:** Extensive experiments conducted on seven real-world public datasets and one real-world dataset tailored for sparse endogenous prediction demonstrate that the TimeSeed model significantly outperforms existing SOTA models in prediction accuracy. Additionally, TimeSeed features fewer parameters (only 0.19M) and low computational cost, making it more amenable to deployment in practical scenarios.

**Summary of the rebuttals**


We have provided detailed and point-by-point responses to all reviewer comments, and we would like to express our sincere gratitude to all reviewers for their valuable inputs. Meanwhile, we greatly appreciate the positive feedback on our work: Reviewer **eg86** expressed recognition of our research; Reviewer **e8px** indicated they would increase their rating once the relevant issues are addressed; Reviewer **b1Px** stated that our responses have sufficiently addressed their core concerns and committed to raising their rating.

* We thank Reviewer **eg86** for acknowledging the novelty of the TimeSeed concept.

  * **Extended Validation of the Robustness of the Evaluation Protocol:** We have added experiments on random missing patterns in Appendix G, analyzed extreme scenarios using only exogenous variables in Appendix I, and systematically evaluated the impacts of reducing the quantity and length of exogenous variables, reducing endogenous variables, and varying historical window lengths in Section 4.5 of the main text. Additionally, we have supplemented error analysis and statistical tests of repeated experiments in Appendix J.

  * **Supplementary Experiments on the "Two-Stage Pipeline Baseline":** Following the suggestion, we have added comparative experiments on the two-stage "imputation + prediction" scheme, using the advanced PatchTST for imputation and PatchTST/TimeXer respectively for prediction.

* We thank Reviewer **e8px** for encouraging us to further improve our work and committing to a rating increase.

  * **Use of Real-World Datasets:** We have added comparative experiments based on the real-world clinical dataset PhysioNet. Its physiological variables (e.g., blood glucose) are naturally sparse, fully aligning with the research scenario of sparse endogenous variables + complete exogenous variables.

  * **Randomness and Statistical Significance:** We have systematically supplemented experiments under random sampling strategies in Appendix G, reported statistical analyses of repeated experiments in Appendix J, and added experiments on the extreme scenario of "using only exogenous variables" in Appendix I.

  * **Impact of Longer Context Windows:** We have systematically analyzed the influence of different lookback window lengths on model performance in Appendix C.

* We thank Reviewer **b1Px** for acknowledging our rebuttal and agreeing to raise the rating.

  * **Real-World Datasets:** We have added comparative experiments based on the real-world clinical dataset PhysioNet. Its physiological variables (e.g., blood glucose) are naturally sparse, fully aligning with the research scenario of sparse endogenous variables + complete exogenous variables.

  * **Rationale for Using Pearson Correlation Coefficient Instead of DTW:** We have re-run the case study code, updated Figure 3 (right), and supplemented the reported results using DTW as the metric.

---

### Meta-Review · Area_Chair_zUAc · 2026-01-02

**Summary:**

This research proposes TimeSeed, a lightweight two-stage framework for time series forecasting under the setting of sparse endogenous variables and dense exogenous variables. The method reconstructs a dense latent representation of the target series using time-domain, frequency-domain, and adaptive multi-scale components, followed by linear forecasting.

The approach is efficient and achieves good empirical performance across several traditional time series benchmarks under the constructed sparsity setting. The authors provided a thorough and well-organized rebuttal, adding additional experiments, clarifications, and comparisons to address reviewer concerns.

Overall, the problem setting in this work is important and practically motivated. However, despite the quality of the rebuttal and incremental experimental improvements, the core technical contributions remain limited, and the evaluation still falls short of convincingly demonstrating broad real-world impact.

**Reviewer Concerns:**

Across the three reviews, the main concerns can be summarized as follows:

+ The methodology largely follows a compositional design, combining established ideas such as time series decomposition and multi-scale forecasting. While thoughtfully integrated, the approach does not introduce a fundamentally new modeling principle, and the contribution is closer to system design than methodological innovation.

+ The evaluation primarily relies on standard (and outdated) forecasting benchmarks with artificially imposed sparsity patterns. While the authors added experiments on a real-world clinical dataset and random missingness during rebuttal, concerns remain that the empirical evidence does not yet sufficiently demonstrate robustness across diverse and realistic scenarios.

+ Reviewers questioned the lack of an explicit reconstruction objective, noting a mismatch between the paper's framing (reconstruction-then-prediction) and the actual training objective. While the authors provided reasonable justification and indirect evidence via ablations and correlations, the interpretability and grounding of the reconstruction stage remain limited.

+ Although the rebuttal added a two-stage imputation+forecasting baseline and improved fairness in comparisons, reviewers still felt that the gains over simpler or modular alternatives were not consistently decisive enough to establish clear methodological advantage.

The rebuttal was excellent in quality and responsiveness. It addressed many clarification-level issues raised by the reviewers, including:
+ Adding real-world data experiments and robustness checks
+ Clarifying experimental protocols and sparsity construction
+ Providing additional ablations and statistical analyses
+ Including additional baselines suggested by reviewers

However the rebuttal does not fundamentally change the overall assessment of this paper's contribution. The central concerns regarding limited technical contributions and insufficiently compelling validation remain only partially addressed. The added experiments strengthen the empirical case but do not fully resolve questions about generality, paradigm shift, or long-term impact relative to prior work.

**Reviewer Scores:**

One reviewer was marginally positive, one reviewer was initially negative and improved their stance after rebuttal, but primarily on clarity and experimental completeness rather than the work's significance. Overall, reviewer opinions converge on recognizing solid engineering and careful experimentation, but diverge on whether the contribution meets the bar for acceptance.

---

### Decision · Program_Chairs · 2026-01-26

Reject